# Purely Agentic Black-Box Optimization for Biological Design

## Abstract

Many key challenges in biological design—such as small-molecule drug discovery, antimicrobial peptide development, and protein engineering—can be framed as black-box optimization over vast, complex structured spaces. Existing methods rely mainly on raw structural data and struggle to exploit the rich scientific literature. While large language models (LLMs) have been added to these pipelines, they have been confined to narrow roles within structure-centered optimizers. We instead cast biological black-box optimization as a fully agentic, language-based reasoning process. We introduce Purely Agentic BLack-box Optimization (`PABLO`), a hierarchical agentic system that uses scientific LLMs pretrained on chemistry and biology literature to generate and iteratively refine biological candidates. On both the standard GuacaMol molecular design and antimicrobial peptide optimization tasks, `PABLO` achieves state-of-the-art performance, substantially improving sample efficiency and final objective values over established baselines. Compared to prior optimization methods that incorporate LLMs, `PABLO` achieves competitive token usage per run despite relying on LLMs throughout the optimization loop. Beyond raw performance, the agentic formulation offers key advantages for realistic design: it naturally incorporates semantic task descriptions, retrieval-augmented domain knowledge, and complex constraints. In follow-up *in vitro* validation, `PABLO`-optimized peptides showed strong activity against drug-resistant pathogens, underscoring the practical potential of `PABLO` for therapeutic discovery.

## 1 Introduction

Searching for novel biological entities, *i.e.*, small molecules, antimicrobial peptides, and proteins, has often been approached as a high-dimensional black-box optimization problem (Negoescu et al., 2011; Gómez-Bombarelli et al., 2018; Jin et al., 2018; Ahn et al., 2020; Tripp et al., 2020; Deshwal & Doppa, 2021; Grosnit et al., 2021; Maus et al., 2022). The objective is to identify discrete sequences or graphs that maximize a costly, often non-differentiable function representing desired properties such as binding affinity, solubility, stability, or antimicrobial potency.

Methods for biological black-box optimization span genetic algorithms (Yoshikawa et al., 2018; Nigam et al., 2021; Jensen, 2019b; Gao et al., 2022b; Nigam et al., 2019; Brown et al., 2019; Fu et al., 2021; Bradshaw et al., 2020), Bayesian optimization (Tripp et al., 2021; Korovina et al., 2020; Moss et al., 2020; Maus et al., 2022; Notin et al., 2021; Lee et al., 2025; Torres et al., 2024; Maus et al., 2023), reinforcement learning (Olivecrona et al., 2017b; Zhou et al., 2019; Ahn et al., 2020), and deep generative modeling (Jin et al., 2018; Bradshaw et al., 2020; Maus et al., 2022; Notin et al., 2021; Lee et al., 2025; Wang et al., 2025b; Szymczak et al., 2023). These approaches typically train generative models such as diffusion models, VAEs, and flow matching models. Because they operate only on domain-specific input modalities (e.g., SMILES/SELFIES strings, amino acid or nucleic acid sequences), they largely ignore the rich human knowledge in natural language—mechanistic insights, structure–activity relationships, and domain heuristics, unless this information is manually engineered into the models or search operators. As a result, they often must relearn key biological principles from scratch for each new task.

This limitation has, of course, inspired a new wave of LLM-enhanced optimization strategies aimed at incorporating higher-level knowledge. A growing body of work integrates LLMs into evolutionary loops as intelligent variation operators (Wang et al., 2025a; Novikov et al., 2025), or embeds them within Bayesian Optimization (BO) pipelines to provide structural priors, candidate generators, or surrogate models (Agarwal et al., 2025; Akke et al., 2025; Liu et al., 2024; Han et al., 2025). Going even further, Yang et al. (2024); Guo et al. (2024); Meindl et al. (2025) treats the LLM *itself* as an optimizer via iterative prompting and feedback, while Pandit et al. (2025) proposes a

[1]Anonymous Institution, Anonymous City, Anonymous Region, Anonymous Country. Correspondence to: Anonymous Author <anon.email@domain.com>.

Preliminary work. Under review by the International Conference on Machine Learning (ICML). Do not distribute.

hybrid LLM-guided optimization system that incorporates language-model reasoning into a classical black-box optimization workflow. The motivation to study how LLMs can be incorporated into design is natural here—a large body of recent work (Bran et al., 2024; Boiko et al., 2023; Ünlü et al., 2025; Kim et al., 2025; Jones et al., 2025) has clearly demonstrated an increasing capability for multimodal reasoning, prediction, and question answering over molecules and language over time.

Given the growing use of LLMs in biological optimization pipelines, it is natural to ask what the extreme case looks like: what if we remove manually designed optimization strategies entirely and rely solely on language models to explore the design space and develop their own strategies? This approach offers clear benefits: (a) LLMs can incorporate domain knowledge acquired during pretraining into the optimization; (b) domain-specific tools can be used via agentic tool calling; and (c) the optimization process becomes highly flexible and easily adaptable to new settings by modifying the LLMs' context.

To this end, we propose Purely Agentic BLack-box Optimization (PABLO), a framework where LLMs pretrained on chemistry and biology literature drive the entire optimization loop. Unlike methods that embed LLMs within BO or evolutionary scaffolds, PABLO coordinates multiple LLM agents with distinct roles to explore the design space. The *Planner Agent* synthesizes high-level search strategies from experimental history, the *Worker Agents* execute these strategies, and the *Explorer Agent* proposes diverse global hypotheses. This factorization is highly generic: the system is agnostic *a priori* about how to construct new candidates—there is no acquisition or fitness function, and all construction strategies are proposed by an LLM. This hierarchy enables online revision of the search strategy, a capability that is difficult to express in LLM-enhanced BO/EA pipelines.

PABLO incorporates scientific knowledge throughout optimization in two complementary ways. First, its foundation models are pretrained on chemistry and biology literature, yielding representations that encode relationships and heuristics beyond what structure-only optimization can provide. Second, the agentic framework supports external tools such as retrieval-augmented literature search (Lewis et al., 2020), enabling targeted access to domain knowledge on demand. PABLO also seamlessly incorporates semantic task descriptions and complex constraints via natural language interactions with the LLM.

Our contributions are summarized as follows:

- We introduce PABLO, a general-purpose black-box optimization algorithm for biological design problems that replaces traditional structure-centric search components with a coordinated set of LLM-driven agents. To our knowledge, PABLO is the first hierarchical, purely agentic framework that formulates biological black-box optimization as end-to-end language-driven reasoning process rather than as an LLM component embedded within a structure-centric optimizer.

- We evaluate PABLO on antimicrobial peptide design and 10 challenging GuacaMol molecular design tasks, showing it consistently outperforms structure-based baselines (Graph GA (Jensen, 2019a), NF-BO (Lee et al., 2025), GEGL (Ahn et al., 2020)) and recent LLM-based approaches (AlphaEvolve (Novikov et al., 2025), LLAMBO (Liu et al., 2024)), finding higher-quality solutions with far fewer black-box evaluations and setting a new state of the art on GuacaMol.

- Somewhat surprisingly, although PABLO relies entirely on LLMs, it does not require substantially more total tokens than existing methods that use LLMs only as a sub-component—and in some cases is significantly more token-efficient.

- We show that expressing the optimization loop in natural language allows integrating auxiliary problem information without redesigning the algorithm. In particular, PABLO can incorporate (i) semantic task descriptions, (ii) external literature via retrieval-augmented tools, (iii) complex design constraints, and (iv) diverse portfolio optimization objectives from Maus et al. (2023).

- We experimentally validate PABLO-designed peptides using *in vitro* minimum inhibitory concentration (MIC) assays, which show significant antimicrobial activity against clinically relevant pathogens.

## 2  Method

Purely Agentic BLack-box Optimization (PABLO) is a hierarchical agentic framework that treats black-box optimization as an iterative loop of global strategy selection followed by local refinement. In each iteration, the local refinement phase improves the current best solutions using strategies chosen at the global level. In contrast, common methods like Bayesian optimization (BO) and evolutionary strategies (EA) fix their candidate selection approach in advance. For instance, BO typically commits to a static acquisition function strategy, with optimization feedback affecting only updates to the surrogate model posterior.

PABLO does not commit to any fixed strategy beyond factoring the optimization problem into the hierarchical agentic system defined below. Instead, global-level agents infer patterns from the observed optimization history and propose new hypotheses online for making progress.

We consider optimization over a structured discrete space $\mathcal{X}$ (*e.g.*, protein/peptide sequences or molecular representations). We describe PABLO in terms of maximization:

$$\underset{x \in \mathcal{X}}{\text{maximize}} \, f(x). \tag{1}$$

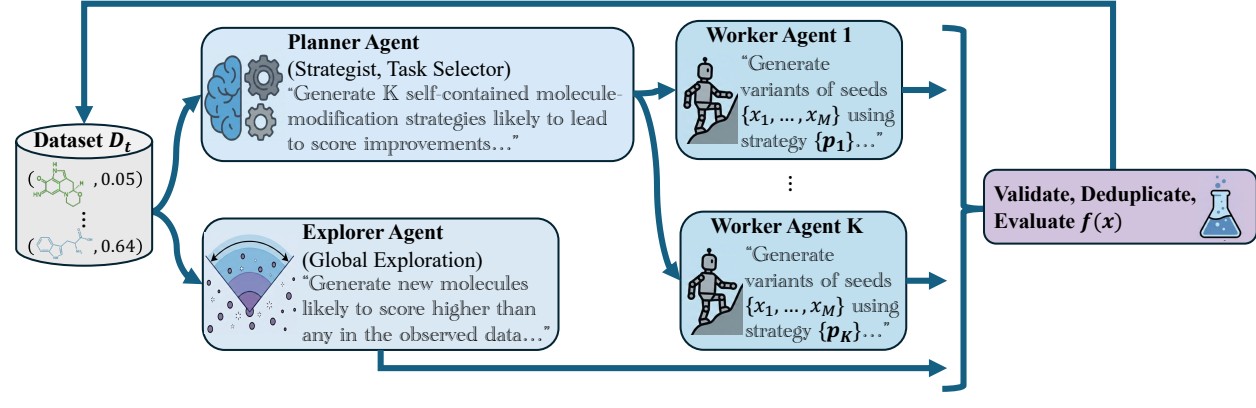

*Figure 1.* A graphical overview of one optimization iteration of PABLO. Each iteration begins with (i) global candidate exploration, (ii) strategy generation via the Planner Agent, and (iii) local refinement of incumbents via planner-proposed strategies. All candidate generations are filtered for validity, novelty, and feasibility before evaluation. PABLO-pseudocode is also provided in Algorithm 1.

We assume a strict oracle evaluation budget $N_{\text{budget}}$. After $t$ oracle calls, the optimization history is $\mathcal{D}_t = \{(x_i, y_i)\}_{i=1}^t$. A design goal of PABLO is *domain generality*: the same optimization loop applies across biological modalities (*e.g.*, peptides, small molecules, or other structured discrete objects). Switching domains mainly requires changing the natural-language domain description (e.g., "SMILES string" $\leftrightarrow$ "peptide sequence") and lightweight validity filters, without modifying the core algorithm.

### 2.1 Distilling Data into Context

LLMs need a compact representation of the current optimization state that reflects both successful and attempted solutions. From the dataset $\mathcal{D}_t$, we construct a straightforward global context $C_{\text{global}}$ to meet these needs:

❶ Include the top-$k$ best candidates as positive exemplars.

❷ Sample additional candidates from the remaining history by sorting them according to their rank and selecting them uniformly with a random initial offset.

Step ❷ ensures the context spans the full performance spectrum. This context includes successful and unsuccessful designs, allowing agents to infer implicit structure–activity relationships and propose edits that meaningfully change candidate scores.

### 2.2 Global Search Strategies

Black-box optimization typically requires (1) **global exploration** to find new high-potential regions of $\mathcal{X}$ and (2) **local exploitation** to iteratively refine candidates once a promising direction is found. We introduce one agent for each role, using chain-of-thought–finetuned models that can analyze the global optimization state and apply scientific knowledge captured during training. We use Intern-S1 (Bai et al., 2025) for both of:

1. **An Explorer Agent** (Global Search). This agent directly proposes new candidates based on the global optimization state and inferred structure–activity trends.

This agent is explicitly encouraged to explore new structures.

2. **A Planner Agent** (Local Strategy Design). This agent maintains a memory of local search tactics and develops new ones from the global state and this memory. The model's scientific understanding helps it propose and prioritize strategies that are more likely to yield functional improvements. Example tactics proposed by the Planner can be found in Section K.

#### 2.2.1 EXPLORER

The Explorer Agent receives the global context $C_{\text{global}}$ and directly proposes a batch of candidates:

$$B_{\text{global}} \leftarrow \text{EXPLORERAGENT}(C_{\text{global}}). \quad (2)$$

See Section H for example Explorer Agent prompts.

The Explorer in isolation most closely resembles how many previous LLM-based optimization methods work: it reviews a summary of the current data and suggests new molecules. It suggests molecules iteratively and receives feedback until it has failed to improve the objective MAX_FAILS times in a row.

#### 2.2.2 PLANNER

Global search can discover promising input regions, but it is inefficient for fine-grained refinement. To focus local edits, we use a second global agent, the Planner. The Planner agent designs and selects local search strategies expressed as natural language *tasks*.

The Planner has access to a **Task Registry** $\mathcal{R}$: a bounded library of task prompts specifying reusable mutation operators or refinement heuristics (*e.g.*, "increase hydrophobicity while preserving motif" for sequences, or "replace unstable functional group" for molecules). Each task in the registry tracks empirical performance statistics:

• attempts: number of times the task has been executed;

- `successes`: number of executions that produced at least one improving candidate;
- `success rate`: successes / attempts.

The registry is initialized with a small set of domain-specific "default" tasks which represent reasonable, simple modification strategies for the given domain, and provide examples that help the `Planner` learn the structure and style of effective task prompts. See Section I.1 for example "default" tasks.

Given $C_{\text{global}}$ and a registry performance summary, the Planner Agent returns a set $\mathcal{P}_{\text{work}}$ of task prompts to execute:

$$\mathcal{P}_{\text{work}} \leftarrow \text{PLANNERAGENT}(C_{\text{global}}, \mathcal{R}). \qquad (3)$$

Each round, the `Planner` is prompted to output "8-10 tasks total" including "2-3 exploitation tasks (targeted at patterns you observed)", "2-3 exploration tasks (creative, untried modification types)", and "2-4 reliable existing tasks that have ($> 0\%$) success rates". See Section I for examples of full `Planner` prompts combining $C_{\text{global}}$ and $\mathcal{R}$.

The registry has a fixed maximum size. When adding a new task would exceed capacity, PABLO prunes the worst-performing non-default task (by success rate with tie-breaking on attempts), ensuring the library remains compact while preserving effective tactics.

### 2.3 Executing Local Search Strategies (Worker)

Each task prompt $p \in \mathcal{P}_{\text{work}}$ the `Planner` generates is given to independent `Worker` agents. The goal of each `Worker` agent is to use the strategy they are provided to locally improve a single "seed input" from the current history $\mathcal{D}_t$—$K$ tasks and $M$ seed inputs, thus resulting in $K \times M$ worker agent runs per optimization round.

**Seed Input Selection**  A set of $M$ seed candidates $S_{\text{seed}} = \{x^{(1)}, \ldots, x^{(M)}\}$ is chosen from $\mathcal{D}_t$ via greedy selection under a diversity threshold, using a domain-specific distance function $\text{dist}(\cdot, \cdot)$ (*e.g.*, normalized edit distance for sequences or fingerprint distance for molecules). This ensures local search is initiated from distinct regions rather than near-duplicates.

**Worker Hill Climbing.**  Each worker aims to optimize its seed using the given strategy until progress plateaus. A `Worker` starts with its seed molecule as $x_{\text{curr}}$ and is repeatedly prompted to improve this molecule with the planner strategy $p$, producing a batch:

$$B \leftarrow \text{WORKERAGENT}(p, x_{\text{curr}}), \qquad (4)$$

See Section J for example `Worker` prompts that combine $p$ and $x_{\text{curr}}$. The batch is validated, deduplicated, and used to update $x_{\text{curr}}$ if any member improves performance. A `Worker` terminates after MAX_FAILS consecutive failures to improve $x_{\text{curr}}$, and the outer loop in Figure 1 ends when

all workers have terminated. Outcomes are recorded in the task registry $\mathcal{R}$, allowing the `Planner` to allocate more budget to productive local strategies.

### 2.4 Validation, Deduplication, and Execution

LLMs may generate candidates that are invalid, duplicates, or violate domain-specific constraints. PABLO post-processes all agent outputs before evaluation. This post-processing step parses candidates in free-form text, enforces domain validity (*e.g.*, alphabet constraints or canonicalization), scores duplicates by memoization, and applies hard feasibility filters.

Biological design problems often require constraints such as synthesizability or similarity to a known template. PABLO supports both soft and hard constraints:

- Soft constraints are encoded in the prompts of the `Explorer` and `Worker` agents to bias generation toward feasible regions;
- Hard constraints are encoded by an indicator function and infeasible candidates are rejected before evaluation.

### 2.5 Diverse Portfolio Optimization

In many biological design pipelines, a single optimum is insufficient due to downstream failure risks (e.g., toxicity or poor manufacturability). Instead, practitioners often require a diverse *portfolio* of many strong candidates (Maus et al., 2023; Torres et al., 2024). PABLO supports this with a straightforward adaptation to optimize an aggregate score over a diverse set:

$$\begin{aligned}
\arg\max_{S \subset \mathcal{X}, |S| = M} \quad & \text{Agg}\left(\{f(x) : x \in S\}\right) \\
\text{subject to} \quad & \text{dist}(x_i, x_j) \geq \beta \quad \forall i \neq j,
\end{aligned} \qquad (5)$$

where Agg is an aggregation function (*e.g.*, the average), and dist is any domain-specific measure of diversity, *e.g.*, sequence edit distance and molecular fingerprint distance.

### 2.6 Extensibility via External Tools.

External tools can be incorporated into PABLO without modifying the core optimization loop (Algorithm 1). Because each agent operates through natural language prompts, we can add capabilities by granting agents access to callable tools (*e.g.*, retrieval systems, property predictors, simulation interfaces) (Schick et al., 2023) that augment reasoning with external knowledge. Tools can be assigned to any agent as needed; for instance, global exploration often benefits from giving the `Explorer` access to knowledge tools. In Section 3.4, we show that a literature retrieval tool improves optimization performance on molecular design tasks, illustrating how domain knowledge can be incorporated.

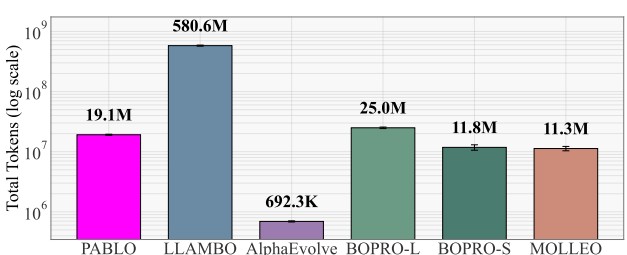

*Figure 2.* Comparing the average number of LLM tokens used per run by PABLO and other LLM-based baselines.

## 3 Experiments

We evaluate PABLO on two biological design domains: (i) small-molecule optimization on GuacaMol benchmark tasks (Brown et al., 2019) and (ii) antimicrobial peptide (AMP) design using a black-box minimum inhibitory concentration (MIC) oracle (Torres et al., 2025). Our experiments are designed to answer: (1) How does the base version of PABLO, without task descriptions or other tools, perform relative to state-of-the-art molecular optimization baselines? (2) How do the different components of PABLO contribute to its final performance? (3) Does PABLO benefit from agentic capabilities such as task descriptions for grey-box optimization and tools like literature search? (4) Can PABLO handle extensions without substantial system modification: constraints and diverse portfolio optimization?

The code and data for reproducing experiments are available at https://anonymous.4open.science/r/agentic-biological-design-753B/.

### 3.1 Experimental Setup

**Small-molecule benchmark tasks.** For molecule optimization, we evaluate on 10 GuacaMol tasks (Brown et al., 2019), selected from among the "challenging" tasks that prior work does not already achieve perfect scores on (1.0). These tasks involve optimizing multi-property objectives over the discrete space of valid SMILES strings.

**AMP design tasks.** For peptides, we optimize antimicrobial peptides (AMPs) to *minimize* the mean predicted MIC (lower is better) against the panel of 11 pathogenic bacteria listed in Table E.1. We use the APEX 1.1 model (Torres et al., 2025), an in-silico predictor of MIC, as our black-box objective function. We report the best mean predicted MIC found so far over the evaluation budget.

**Baselines.** We compare PABLO to strong baselines appropriate for each domain and setting. In all plots, PABLO and all baseline methods show the mean and standard error over 10 runs unless otherwise noted.

- **GuacaMol Baselines.** We compare to recent state-of-the-art methods from Bayesian optimization (NF-BO (Lee et al., 2025)), reinforcement learning (GEGL (Ahn et al., 2020)), and LLM-enhanced

optimization (AlphaEvolve (Novikov et al., 2025), LLAMBO (Liu et al., 2024), MOLLEO (Wang et al., 2025a), BOPRO (Agarwal et al., 2025)). For BOPRO, BOPRO-L and BOPRO-S signify runs with the Mistral-Large and Mistral-Small LLMs respectively (Mistral AI, 2024; 2025). Since BOPRO-L expensive ( 600 USD per task), we only run BOPRO-L on 3/10 tasks. We also compare to "Random": random sampling from the SELFIES-VAE (Maus et al., 2022). Additionally, in Table 1, we compare against 25 additional baseline methods curated by the recent GuacaMol benchmarking study of Gao et al. (2022a), and refer readers there for full citations and discussion of each method.

- **AMP Baselines.** We compare to APEX-GO (Torres et al., 2024) (latent-space Bayesian optimization), HydrAMP (Szymczak et al., 2023) (AMP generation with a conditional variational autoencoder (VAE)), and PepDiffusion (Wang et al., 2025b) (AMP generation with latent diffusion). (See Figure 5.)

**Implementation Details.** Across all experiments, PABLO optimizes under a strict evaluation budget of 20,000 oracle calls. Each evaluation corresponds to one valid candidate design (SMILES string or peptide sequence) after filtering for validity and novelty. Further implementation details are provided in Section D.

LLM outputs undergo deterministic post-processing before oracle calls: candidates are parsed from raw text generations, filtered for domain validity (e.g., amino-acid alphabet constraints, SMILES canonicalization). To ensure the oracle budget is not spent on infeasible or duplicate designs, duplicates are removed (both within-batch and against history), and hard constraints are enforced via rejection. When comparing to prior work in Figure 3, PABLO **does not** access task descriptions or literature search tools; models see only molecules and their numeric scores.

### 3.2 Molecule Optimization on GuacaMol

Figure 3 shows optimization trajectories on 10 GuacaMol tasks. PABLO improves faster and more consistently reaches higher final objective values than all baselines, and is clearly outperformed on only one task (adip). Figure 2 compares the average LLM tokens used per run for all LLM-based methods. AlphaEvolve uses very few LLM tokens overall, calling an LLM only occasionally to tweak its candidate-generating program. Among methods that use LLMs in the loop, PABLO's token usage is roughly average, despite relying on LLMs exclusively.

**Aggregate performance and literature comparison.** To contextualize these results against a broader benchmark suite, Table 1 reports Top-1 performance at 10K evaluations and compares PABLO against 26 additional literature baselines reported by Gao et al. (2022a). Compared against

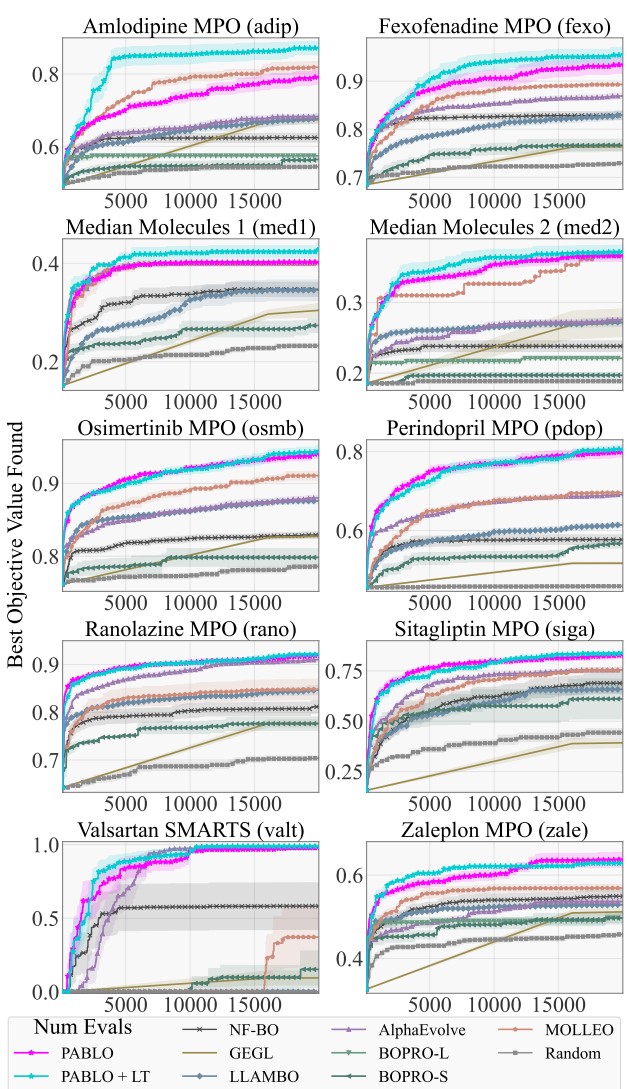

*Figure 3.* GuacaMol optimization results on 10 tasks. Curves show objective value of the best molecule found so far as a function of black-box evaluations. `PABLO` achieves state-of-the-art performance, rapidly reaching strong objective values across tasks.

these baselines, `PABLO` ranks first in every task, often by substantial margins.

### 3.3 `PABLO` Extensions

In Figure B.2 and Table B.8, we study two optional extensions to PABLO: a Literature Retrieval Tool (LT) and Task Awareness (TA).

**Extension 1: Literature Tool (LT).** We equip the `Explorer` with a literature retrieval tool (LT) that implements a retrieval-augmented generation (RAG) pipeline: given a single molecule (a SMILES string chosen by the agent), it returns a structured dictionary of context from papers that either explicitly mention the queried molecule or highly similar molecules found via fingerprint similarity search. Examples of queries and retrieved context are in

*Table 1.* Benchmark on 10 GuacaMol tasks. We report the mean and standard deviation of top-1 molecules from 5 independent runs at a 10K evaluation budget following Table 20 in Gao et al. (2022a). We compare 26 methods in total and show the top 3 below, with the full table in Appendix B.

| Task | `PABLO` | Graph GA | REINVENT |
|------|---------|----------|----------|
| med1 | $0.398_{\pm0.034}$ | $0.350_{\pm0.050}$ | $\mathbf{0.399}_{\pm\mathbf{0.058}}$ |
| med2 | $\mathbf{0.339}_{\pm\mathbf{0.035}}$ | $0.324_{\pm0.040}$ | $0.332_{\pm0.045}$ |
| pdop | $\mathbf{0.772}_{\pm\mathbf{0.041}}$ | $0.625_{\pm0.054}$ | $0.644_{\pm0.071}$ |
| osmb | $\mathbf{0.919}_{\pm\mathbf{0.017}}$ | $0.880_{\pm0.029}$ | $0.909_{\pm0.040}$ |
| adip | $\mathbf{0.853}_{\pm\mathbf{0.080}}$ | $0.783_{\pm0.078}$ | $0.735_{\pm0.086}$ |
| siga | $\mathbf{0.773}_{\pm\mathbf{0.043}}$ | $0.689_{\pm0.214}$ | $0.080_{\pm0.034}$ |
| zale | $\mathbf{0.606}_{\pm\mathbf{0.011}}$ | $0.421_{\pm0.086}$ | $0.478_{\pm0.150}$ |
| valt | $\mathbf{0.951}_{\pm\mathbf{0.079}}$ | $0.000_{\pm0.000}$ | $0.197_{\pm0.382}$ |
| rano | $\mathbf{0.893}_{\pm\mathbf{0.015}}$ | $0.810_{\pm0.072}$ | $0.865_{\pm0.068}$ |
| fexo | $\mathbf{0.934}_{\pm\mathbf{0.045}}$ | $0.845_{\pm0.053}$ | $0.910_{\pm0.073}$ |
| **Sum** | **7.439** | 5.727 | 5.549 |
| **Rank** | 1 | 2 | 3 |

Figure D.1. We do not modify the architecture of `PABLO`; the tool is simply provided as an additional resource. As shown in Figure 3 (`PABLO + LT` curves) and Table B.8, enabling LT yields clear improvements on several tasks.

**Extension 2: Task Awareness (TA).** Task Awareness (TA) gives agents an explicit natural-language description of the optimization objective, which we append to the prompts of both the `Explorer` and `Planner`. For example, the Amlodipine MPO task is: *"Maximize similarity to amlodipine while having exactly 3 total rings."* As shown in Figure B.2 and Table B.8, TA makes many Guacamol optimization tasks nearly trivial: Intern S1 can directly propose structures aligned with the scoring logic, often reaching near-optimal solutions within a few hundred function evaluations. Because of this, all primary PABLO results in Figure 3, Table 1, etc. are reported *without* task descriptions, matching the standard black-box benchmark assumption. `+ TA` is only used where explicitly indicated.

### 3.4 `PABLO` Ablations

We ablate the major components of `PABLO` here–see Section B.2 for additional studies of `PABLO`-hyperparameters.

In Figure 4, we ablate major components of `PABLO` on representative GuacaMol MPO tasks. Removing the `Explorer` (*PABLO w/o Explorer*) consistently degrades performance, showing that global, context-driven exploration is crucial for rapidly locating promising regions. We also ablate the `Planner` by replacing it with fixed `Worker` prompts: (i) *3 Static Worker Prompts* from the default Task Registry and (ii) *10 Static Worker Prompts*, which add seven hand-designed molecule-editing prompts. These static-prompt baselines are competitive, but `PABLO` with the Planner Agent consistently achieves higher final perfor-

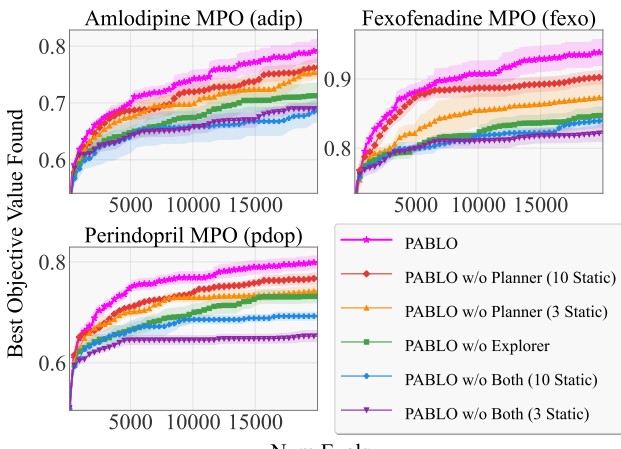

*Figure 4.* PABLO ablation on representative GuacaMol tasks showing the contribution of the Planner and Explorer Agents.

mance and better sample efficiency. The Planner Agent even outperforms the 10-prompt baseline, showing that `Planner` prompts are more effective than a fixed set. The static prompts are listed in Section L. Removing *both* the `Explorer` and `Planner` (*PABLO w/o Both*) gives the worst performance, underscoring their complementarity.

### 3.5 Peptide Optimization (AMP Design)

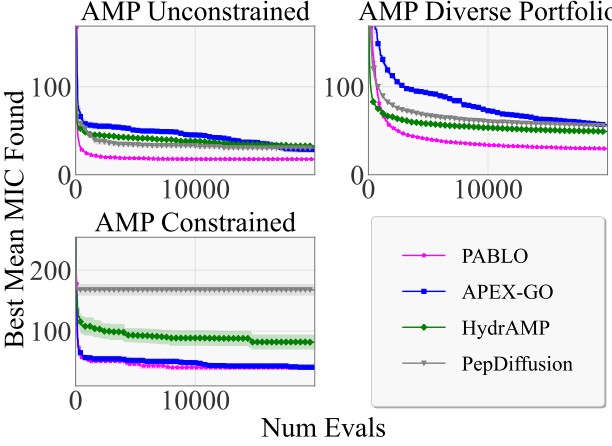

*Figure 5.* Antimicrobial peptide (AMP) optimization. We plot predicted MIC versus black-box evaluations (lower MIC is better). (**Upper Left**): The predicted MIC of the best peptide found so far. (**Lower Left**): Template-constrained optimization; we show the predicted MIC of the best "feasible" peptide found so far. (**Upper Right**): Template-free optimization of a diverse portfolio of 20 AMPs; we show the mean predicted MIC of the best sufficiently-diverse portfolio so far.

Figure 5 Upper Left Panel shows AMP optimization. PABLO improves rapidly and consistently identifies peptides with substantially lower predicted MIC than all baselines under the same evaluation budget.

### 3.6 Constraints and Diverse Portfolio Optimization

**Template-constrained AMP optimization.** We evaluate PABLO under a realistic hard constraint: candidates must be at least 75% similar to one of 10 "trusted" template peptides, simulating lead optimization rather than de novo design. These 10 templates, mined from extinct organisms and selected by Torres et al. (2025), were chosen because extinct-like peptides may better evade antibiotic resistance in modern bacteria. As shown in Figure 5 (Lower Left), PABLO maintains its advantage under these template constraints, indicating that prompt modifications and rejection are moderately sufficient to handle constraints.

**Diverse portfolio optimization.** We consider portfolio optimization—targeting simultaneous optimization of 20 peptides—to reduce downstream failure risk when optimizing against an *in silico* predictor. We impose a pairwise diversity constraint requiring all pairs in the 20-AMP portfolio to have at least 0.75 dissimilarity by edit distance. Figure 5 (upper right) shows that PABLO substantially improves portfolio quality (mean MIC over the best diverse set) compared to baselines under the same budget.

***In Vitro* Performance.** To validate that a diverse portfolio might lead to strong AMPs, we performed *in vitro* experiments on a portfolio of peptides produced by one run of PABLO. We refer to this portfolio as P1–P20 (sequences and APEX 1.1–predicted MICs are listed in Table B.7). *In vitro* experimental procedures are detailed in Section F.

Figure 6 presents measured *in vitro* MIC values against the 11 target bacteria (B1–B11; see Table E.1) for the three best-performing peptides in the portfolio, ranked by lowest average *in vitro* MIC across B1–B11. Full *in vitro* results for all 20 peptides (P1–P20) are provided in Figure B.1.

Although all 20 peptides had APEX 1.1–predicted MICs below $30\,\mu\mathrm{mol}\,\mathrm{L}^{-1}$, several showed weak or no detectable experimental activity (gray cells in Figure B.1), illustrating the known gap between prediction and biological performance. Nonetheless, for every target species, at least one peptide achieved strong inhibition (MIC $\leq 16\,\mu\mathrm{mol}\,\mathrm{L}^{-1}$).

**Held-out bacteria.** The full heatmap in Figure B.1 also shows in vitro results for nine additional bacteria (B12–B20; see Table E.2) that were not part of the optimization objective. Several peptides still showed strong activity against these held-out targets, indicating broad-spectrum activity.

## 4 Related Work

**Biological Design as Black-Box Optimization.** The discovery of novel biological entities—such as small molecules, peptides, or proteins—can be formalized as a maximization problem of some objective function over a discrete structured space $\mathcal{X}$ (Coley et al., 2020). We seek

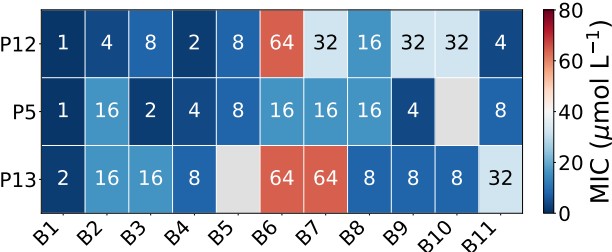

*Figure 6. In vitro* MIC results against the 11 target bacteria (B1–B11; see Table E.1) achieved by the three best-performing peptides from the $M = 20$-peptide portfolio produced by one run of PABLO on the AMP design task. Peptide sequences are listed in Table B.7. For complete *in vitro* results see Figure B.1.

$x^* = \operatorname{argmax}_{x \in \mathcal{X}} f(x)$, where $f$ might measure binding affinity, solubility, or drug-likeness. The machine learning literature offers a diverse set of strategies to solve these biological optimization problems:

**Virtual Screening:** High-throughput approaches evaluate fixed libraries of compounds. Recent methods like MolPAL (Graff et al., 2021b) and active learning strategies (Graff et al., 2021a) iterate between screening and model updating to identify top candidates from pool-based libraries.

**Genetic Algorithms (GAs):** GAs explore $\mathcal{X}$ via stochastic crossover and mutation. These include string-based methods like SMILES-GA (Yoshikawa et al., 2018) and STONED (Nigam et al., 2021), as well as graph-based approaches like Graph GA (Jensen, 2019b). Recent work has also explored augmenting GAs with neural networks (Nigam et al., 2019) or combining them with synthesis trees (Gao et al., 2022b).

**Bayesian Optimization (BO):** BO builds a probabilistic surrogate model to guide acquisition (Frazier, 2018; Shahriari et al., 2015; Garnett, 2023). To handle discrete biological data, recent methods operate either directly over string spaces (e.g., BOSS (Moss et al., 2020)) or over continuous latent spaces learned by generative models (Latent Space BO) (Tripp et al., 2020; Deshwal & Doppa, 2021; Stanton et al., 2022; Lee et al., 2023; Notin et al., 2021; Griffiths & Hernández-Lobato, 2020; Lu et al., 2018; Kusner et al., 2017; Maus et al., 2022; 2023; Lee et al., 2025).

**Deep Generative Models (DGMs):** These models learn a distribution over valid structures. Variational Autoencoders (VAEs) map discrete structures to continuous latent spaces for optimization, including the seminal SMILES-VAE (Gómez-Bombarelli et al., 2018), Junction Tree VAE (Jin et al., 2018), and SELFIES-based models (Maus et al., 2022). Recent advances also include Normalizing Flows (Lee et al., 2025), Diffusion Models (Ho & Salimans, 2021), and score-based modeling like MARS (Xie et al., 2021) and GFlowNets (Bengio et al., 2023).

**Reinforcement Learning (RL)** RL agents learn policies to construct molecules step-by-step, as seen in methods like REINVENT, MolDQN, and GEGL (Olivecrona et al.,

2017b;a; Zhou et al., 2019; Ahn et al., 2020).

**LLM-Enhanced Optimization.** A growing body of work integrates Large Language Models (LLMs) into optimization pipelines. Evolutionary methods such as AlphaEvolve (Novikov et al., 2025) and related approaches (Wang et al., 2025a) use LLMs as intelligent variation operators. Recent BO methods use LLMs to improve surrogate models or candidate generation by providing structural priors (Liu et al., 2024) or proposing candidates for acquisition-driven search (Agarwal et al., 2025; Akke et al., 2025; Han et al., 2025).

Many works study *LLMs as optimizers*, using iterative prompting and textual feedback loops to optimize black-box objectives (Yang et al., 2024; Guo et al., 2024; Meindl et al., 2025). They are usually tested on prompt optimization, math problems, or generic black-box functions, not biological design tasks with structured chemical or sequence spaces. Related work couples LLM reasoning with standard optimizer infrastructure for LLM-guided black-box optimization (Pandit et al., 2025), but typically embeds the LLM in a fixed scaffold (e.g., replacing a random mutation operator) or simple iterative proposal loops without explicit hierarchical strategy roles, rather than giving it autonomy to plan and revise the search strategy. Recent chemistry-focused agent frameworks show the promise of tool-using, stepwise-planning LLM agents for multi-step planning and iterative refinement (Bran et al., 2024; Boiko et al., 2023; Ünlü et al., 2025; Zou et al., 2025; Ramos et al., 2025). We extend this paradigm to *black-box optimization for design*.

## 5  Discussion

Our results demonstrate that a relatively simple agentic factorization of black-box optimization can achieve state-of-the-art performance on standard molecular design benchmarks, outperforming even fairly complex state-of-the-art baselines. The success of PABLO suggests that LLMs are becoming sufficiently knowledgeable and capable of step-by-step reasoning, making them competitive with manually-defined optimization policies. Moreover, this approach is compelling because it easily enables capabilities that can be difficult to incorporate into more purpose-built methods:

- Semantic Task Awareness: Using natural language descriptions of the objective to accelerate the search.
- External Knowledge Integration: Seamlessly incorporating retrieval-augmented generation (RAG) tools to ground optimization in scientific literature.
- Flexible Constraint Handling: Addressing complex, non-differentiable constraints (like peptide template similarity) through simple prompt engineering.

## Impact Statement

This paper presents work whose goal is to advance the field of Machine Learning. There are many potential societal consequences of our work, none which we feel must be specifically highlighted here.

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

# A `PABLO` Pseudocode Algorithm

In Algorithm 1, we provide a pseudocode algorithm for `PABLO`.

---

**Algorithm 1** Purely Agentic BLack-box Optimization (`PABLO`): hierarchical agentic optimization with success persistence and task memory. Each outer iteration alternates global exploration, strategy selection, and local multi-trajectory refinement. All LLM-generated candidates are validated and deduplicated prior to oracle evaluation to avoid wasting budget. In particular, the "Filter" step filters out all candidates that are already in $\mathcal{D}_t$ (duplicates) or violate domain constraints (invalid).

---

1: **Input:** oracle $f$, budget $N_{\text{budget}}$, history $\mathcal{D}_0$, Task Registry $\mathcal{R}$
2: **while** $|\mathcal{D}_t| < N_{\text{budget}}$ **do**
3:     *// Build global context*
4:     $C_{\text{global}} \leftarrow$ HISTORYCOVERAGESAMPLE$(\mathcal{D}_t)$
5:     *// Phase I: Global search with persistence*
6:     fails $\leftarrow 0$
7:     **while** $fails <$ MAX_FAILS **do**
8:         $B_{\text{global}} \leftarrow$ EXPLORERAGENT$(C_{\text{global}})$
9:         $\tilde{B}_{\text{global}} \leftarrow$ FILTER$(B_{\text{global}}, \mathcal{D}_t)$
10:         Evaluate $f(x)$ for $x \in \tilde{B}_{\text{global}}$ and add to $\mathcal{D}_t$
11:         **if** IMPROVEDSTATISTIC$(\mathcal{D}_t)$ **then**
12:             fails $\leftarrow 0$
13:         **else**
14:             fails $\leftarrow$ fails $+ 1$
15:         **end if**
16:         $C_{\text{global}} \leftarrow$ HISTORYCOVERAGESAMPLE$(\mathcal{D}_t)$
17:     **end while**
18:     *// Phase II: Planner Agent selects tasks*
19:     $\mathcal{P}_{\text{work}} \leftarrow$ PLANNERAGENT$(C_{\text{global}}, \mathcal{R})$
20:     *// Phase III: Local multi-trajectory refinement*
21:     **for** each task prompt $p \in \mathcal{P}_{\text{work}}$ **do**
22:         $S_{\text{seed}} \leftarrow$ SELECTDIVERSESEEDS$(\mathcal{D}_t)$
23:         **for** each seed $x^{(j)} \in S_{\text{seed}}$ **do**
24:             $x_{\text{curr}} \leftarrow x^{(j)}$; fails $\leftarrow 0$
25:             **while** fails $<$ MAX_FAILS **do**
26:                 $B_{\text{loc}} \leftarrow$ WORKERAGENT$(p, x_{\text{curr}})$
27:                 $\tilde{B}_{\text{loc}} \leftarrow$ FILTER$(B_{\text{loc}}, \mathcal{D}_t)$
28:                 Evaluate $f(x)$ for $x \in \tilde{B}_{\text{loc}}$ and add to $\mathcal{D}_t$
29:                 **if** IMPROVESTRAJWITHOUTCOLLAPSE$(\tilde{B}_{\text{loc}})$ **then**
30:                     $x_{\text{curr}} \leftarrow \arg\max_{x \in \tilde{B}_{\text{loc}}} f(x)$
31:                     fails $\leftarrow 0$; record success in $\mathcal{R}$
32:                 **else**
33:                     fails $\leftarrow$ fails $+ 1$; record failure in $\mathcal{R}$
34:                 **end if**
35:             **end while**
36:         **end for**
37:     **end for**
38: **end while**

---

## B  Additional Experimental Results

In Table B.1-Table B.6, we provide the remaining columns of Table 1 from the main text.

*Table B.1.* Table 1 Continued

| Task | REINVENT SELFIES | LSTM HC | STONED | GP BO |
|------|------------------|---------|--------|-------|
| med1 | 0.399±0.063 | 0.388±0.064 | 0.295±0.036 | 0.345±0.044 |
| med2 | 0.313±0.040 | 0.339±0.049 | 0.265±0.038 | 0.337±0.033 |
| pdop | 0.610±0.070 | 0.568±0.037 | 0.522±0.027 | 0.562±0.036 |
| osmb | 0.878±0.028 | 0.859±0.023 | 0.848±0.024 | 0.837±0.020 |
| adip | 0.706±0.068 | 0.739±0.063 | 0.638±0.054 | 0.681±0.067 |
| siga | 0.409±0.170 | 0.262±0.079 | 0.526±0.169 | 0.318±0.117 |
| zale | 0.441±0.109 | 0.413±0.126 | 0.373±0.088 | 0.269±0.084 |
| valt | 0.000±0.000 | 0.000±0.000 | 0.000±0.000 | 0.000±0.000 |
| rano | 0.851±0.095 | 0.824±0.073 | 0.862±0.113 | 0.817±0.080 |
| fexo | 0.842±0.044 | 0.818±0.047 | 0.851±0.058 | 0.805±0.053 |
| **Sum** | 5.449 | 5.210 | 5.180 | 4.971 |
| **Rank** | 4 | 5 | 6 | 7 |

*Table B.2.* (Continued)

| Task | DoG-Gen | LSTM HC SELFIES | SMILES GA | SynNet |
|------|---------|-----------------|-----------|--------|
| med1 | 0.322±0.053 | 0.362±0.058 | 0.207±0.014 | 0.244±0.019 |
| med2 | 0.297±0.040 | 0.274±0.031 | 0.210±0.009 | 0.259±0.016 |
| pdop | 0.587±0.044 | 0.521±0.028 | 0.459±0.014 | 0.610±0.039 |
| osmb | 0.850±0.028 | 0.832±0.018 | 0.835±0.019 | 0.821±0.016 |
| adip | 0.621±0.034 | 0.600±0.012 | 0.570±0.006 | 0.596±0.020 |
| siga | 0.252±0.099 | 0.349±0.089 | 0.504±0.145 | 0.067±0.040 |
| zale | 0.343±0.111 | 0.360±0.093 | 0.396±0.097 | 0.402±0.059 |
| valt | 0.000±0.000 | 0.000±0.000 | 0.000±0.000 | 0.000±0.000 |
| rano | 0.823±0.057 | 0.795±0.099 | 0.780±0.082 | 0.783±0.038 |
| fexo | 0.808±0.036 | 0.769±0.039 | 0.771±0.041 | 0.797±0.031 |
| **Sum** | 4.903 | 4.862 | 4.732 | 4.579 |
| **Rank** | 8 | 9 | 10 | 11 |

*Table B.3.* (Continued)

| Task | MIMOSA | DST | GA+D | VAE BO SELFIES |
|------|--------|-----|------|----------------|
| med1 | 0.296±0.039 | 0.281±0.036 | 0.219±0.037 | 0.231±0.017 |
| med2 | 0.238±0.016 | 0.201±0.024 | 0.161±0.028 | 0.206±0.006 |
| pdop | 0.557±0.047 | 0.502±0.026 | 0.337±0.147 | 0.482±0.024 |
| osmb | 0.817±0.022 | 0.827±0.018 | 0.784±0.129 | 0.802±0.010 |
| adip | 0.594±0.009 | 0.582±0.054 | 0.527±0.124 | 0.593±0.022 |
| siga | 0.209±0.085 | 0.205±0.106 | 0.482±0.175 | 0.244±0.083 |
| zale | 0.287±0.103 | 0.344±0.119 | 0.359±0.119 | 0.379±0.091 |
| valt | 0.000±0.000 | 0.000±0.000 | 0.000±0.000 | 0.064±0.072 |
| rano | 0.773±0.139 | 0.752±0.163 | 0.775±0.244 | 0.564±0.065 |
| fexo | 0.743±0.030 | 0.778±0.041 | 0.737±0.174 | 0.707±0.011 |
| **Sum** | 4.514 | 4.472 | 4.381 | 4.272 |
| **Rank** | 12 | 13 | 14 | 15 |

*Table B.4.* (Continued)

| Task | MolPAL | MARS | Screening | VAE BO SMILES |
|------|--------|------|-----------|---------------|
| med1 | 0.309±0.028 | 0.233±0.017 | 0.271±0.029 | 0.267±0.043 |
| med2 | 0.273±0.021 | 0.203±0.015 | 0.244±0.021 | 0.222±0.011 |
| pdop | 0.504±0.020 | 0.488±0.016 | 0.500±0.028 | 0.484±0.028 |
| osmb | 0.816±0.020 | 0.809±0.021 | 0.801±0.016 | 0.801±0.010 |
| adip | 0.651±0.043 | 0.546±0.034 | 0.613±0.039 | 0.611±0.036 |
| siga | 0.117±0.030 | 0.083±0.037 | 0.142±0.060 | 0.114±0.068 |
| zale | 0.286±0.064 | 0.296±0.023 | 0.280±0.101 | 0.139±0.046 |
| valt | 0.000±0.000 | 0.000±0.000 | 0.000±0.000 | 0.064±0.077 |
| rano | 0.556±0.064 | 0.776±0.050 | 0.532±0.059 | 0.598±0.076 |
| fexo | 0.709±0.006 | 0.755±0.034 | 0.706±0.021 | 0.719±0.016 |
| **Sum** | 4.221 | 4.189 | 4.089 | 4.019 |
| **Rank** | 16 | 17 | 18 | 19 |

*Table B.5.* (Continued)

| Task | JT-VAE BO | Pasithea | DoG-AE | GFlowNet |
|------|-----------|----------|--------|----------|
| med1 | 0.212±0.019 | 0.216±0.021 | 0.203±0.014 | 0.237±0.019 |
| med2 | 0.192±0.003 | 0.194±0.006 | 0.201±0.010 | 0.198±0.009 |
| pdop | 0.463±0.019 | 0.447±0.016 | 0.464±0.026 | 0.478±0.021 |
| osmb | 0.800±0.011 | 0.792±0.009 | 0.793±0.026 | 0.817±0.016 |
| adip | 0.585±0.000 | 0.585±0.000 | 0.539±0.017 | 0.482±0.016 |
| siga | 0.169±0.096 | 0.230±0.085 | 0.039±0.033 | 0.045±0.020 |
| zale | 0.302±0.089 | 0.243±0.084 | 0.156±0.093 | 0.118±0.061 |
| valt | 0.000±0.000 | 0.064±0.126 | 0.000±0.000 | 0.000±0.000 |
| rano | 0.587±0.041 | 0.443±0.054 | 0.744±0.025 | 0.701±0.030 |
| fexo | 0.702±0.016 | 0.707±0.041 | 0.723±0.045 | 0.727±0.017 |
| **Sum** | 4.012 | 3.921 | 3.862 | 3.803 |
| **Rank** | 20 | 21 | 22 | 23 |

## B.1 Additional *In Vitro* Results

This section provides complete *in vitro* validation results for the diverse portfolio of $M = 20$ antimicrobial peptides produced by one run of `PABLO`. Experimental procedures for measuring *in vitro* MIC values are detailed in Section F.

Table B.7 lists the amino acid sequences of all 20 optimized peptides (P1–P20) along with their APEX 1.1–predicted MIC

*Table B.6.* (Continued)

| Task | GFlowNet-AL | Graph MCTS | MolDQN |
|------|-------------|------------|--------|
| med1 | 0.229±0.012 | 0.242±0.023 | 0.188±0.028 |
| med2 | 0.191±0.009 | 0.148±0.010 | 0.108±0.009 |
| pdop | 0.464±0.020 | 0.334±0.038 | 0.282±0.062 |
| osmb | 0.812±0.010 | 0.738±0.018 | 0.699±0.018 |
| adip | 0.466±0.016 | 0.483±0.024 | 0.383±0.033 |
| siga | 0.028±0.017 | 0.210±0.088 | 0.015±0.009 |
| zale | 0.048±0.020 | 0.166±0.065 | 0.042±0.024 |
| valt | 0.000±0.000 | 0.000±0.000 | 0.000±0.000 |
| rano | 0.705±0.034 | 0.369±0.096 | 0.171±0.077 |
| fexo | 0.732±0.015 | 0.611±0.024 | 0.532±0.039 |
| **Sum** | 3.675 | 3.301 | 2.420 |
| **Rank** | 24 | 25 | 26 |

values. These peptides were optimized *in silico* to minimize average predicted MIC across 11 target bacteria (B1–B11; species names provided in Table E.1). A key motivation for optimizing a diverse portfolio rather than a single candidate is to mitigate downstream failure risk: *in silico* predictions are imperfect, and peptides may fail to synthesize or exhibit unexpected behavior in solution. By generating 20 diverse candidates that all exhibit strong predicted inhibitory activity, we increase the probability that at least one will demonstrate true efficacy *in vitro* against each target bacterium.

Figure B.1 presents measured *in vitro* MIC values for all 20 peptides against 20 bacterial targets (B1–B20). These include the 11 bacteria used as optimization targets during *in silico* design (B1–B11; see Table E.1) as well as 9 additional bacteria (B12–B20; see Table E.2) that were not part of the optimization objective. The additional bacteria provide an evaluation of broader-spectrum antimicrobial activity beyond the original targets. Gray cells indicate no detectable inhibitory activity at the concentrations tested.

For the original 11 optimization targets (B1–B11), at least one peptide in the portfolio achieved very strong *in vitro* activity (MIC $\leq 16\,\mu\mathrm{mol\,L}^{-1}$) against each bacterium, validating the diverse portfolio optimization strategy. Notably, several peptides also exhibited strong inhibitory activity against the held-out bacteria B12–B20, despite these targets not being included in the optimization objective. This demonstrates that optimizing for efficacy against a diverse panel of bacteria can yield peptides with broad-spectrum antimicrobial properties that generalize to previously unseen targets.

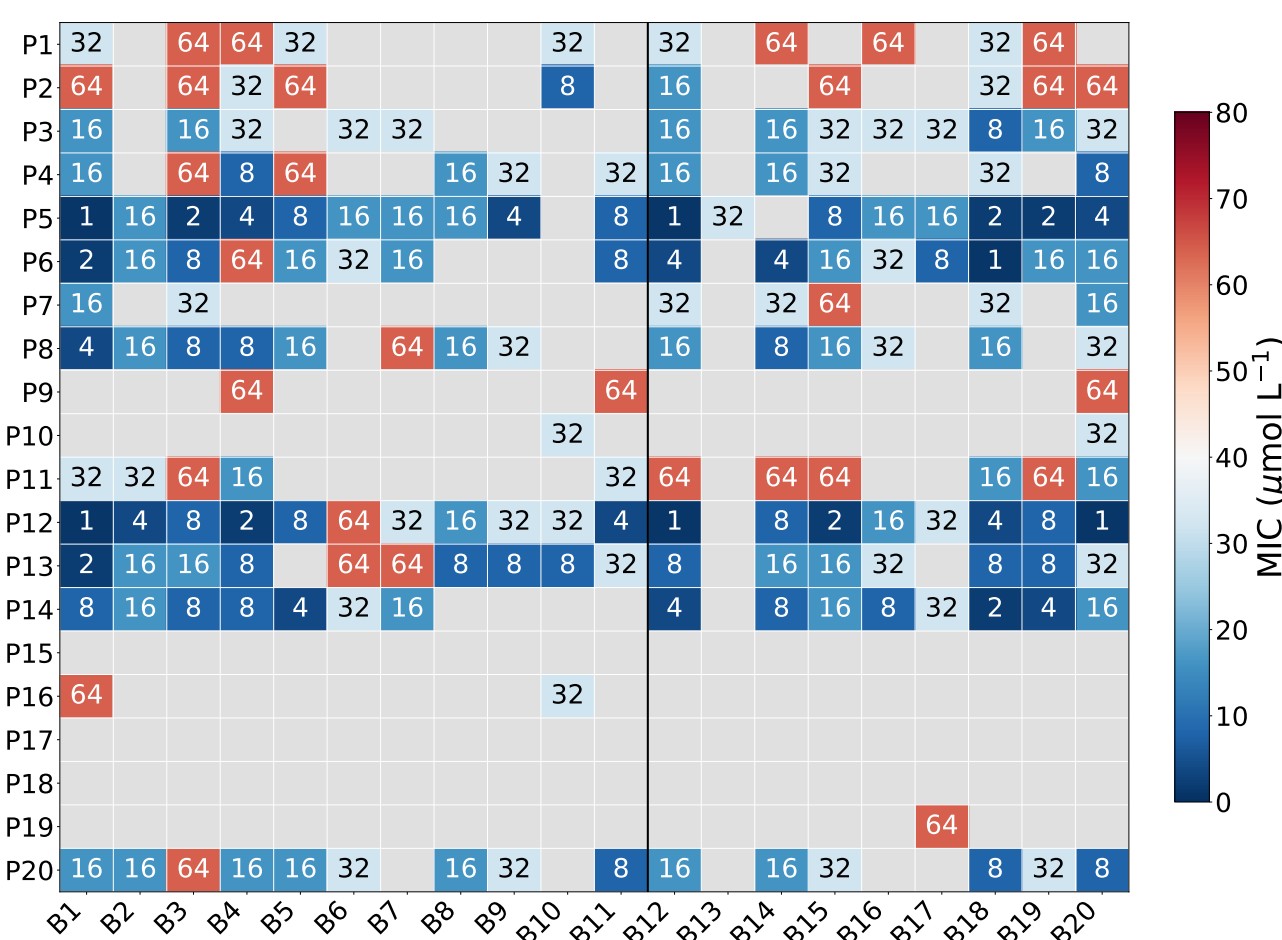

*Figure B.1. In vitro* MIC results for all 20 peptides (P1–P20) in the optimized diverse portfolio produced by one run of `PABLO`. Results are shown against the 11 optimization target bacteria (B1–B11; see Table E.1) as well as 9 additional bacteria (B12–B20; see Table E.2) that were not part of the optimization objective. Peptide sequences for P1–P20 are listed in Table B.7. Gray cells indicate no detectable activity at the tested concentrations.

*Table B.7.* Example portfolio $M = 20$ diverse peptides optimized by a single random run of PABLO. Specifically, this portfolio was the one selected for *in vitro* validation. Peptide IDs P1-P20 used to identify the $M = 20$ unique peptides in the portfolio in all *in vitro* results provided in this paper (e.g., in Figure B.1 and Figure 6). The "Pred. MIC" column provides the APEX 1.1 Model's predicted MICs averaged across the 11 target bacteria listed in Table E.1 (the black-box objective value used during optimization with PABLO).

| Peptide ID | Sequence | Pred. MIC |
|---|:---:|:---:|
| P1 | FLRWKLRFRIRLIL | 17.5 |
| P2 | KIRWRIRILFRLLLKKF | 22.0 |
| P3 | WKRLFKRIKIVLRWF | 23.7 |
| P4 | WRLIILRAARWLLK | 24.7 |
| P5 | IALRRWIFKLAKALKW | 25.2 |
| P6 | RIKFWKLRIIKFF | 25.6 |
| P7 | LKMKIALLKLVAGKKL | 26.2 |
| P8 | KIILKIRWRWLLNIAKLAAFK | 26.6 |
| P9 | FKLWRRWWWLVLR | 26.8 |
| P10 | WRWLAKIAIRAFWKLKIKW | 26.9 |
| P11 | LIRFRFRLKWRLF | 27.0 |
| P12 | KWIKLVRWFKWIKF | 27.1 |
| P13 | IFRWLKRWVFRW | 27.5 |
| P14 | KWKKKIFLKVRFW | 27.7 |
| P15 | FWWKFIRWLRRILLRRFFRW | 27.8 |
| P16 | LKKIWLRIIIKRFLRWKFRLLL | 27.8 |
| P17 | LFWKKFRIRWRWWWLRIFLRWRWLLRWWWILLRFRFF | 28.1 |
| P18 | IRWFKRLRFRLWWWFRFLRRVF | 28.1 |
| P19 | ILKIRFRWKIRLFFKLLRKWLWWF | 28.3 |
| P20 | KIFWRILILGRLLIKRFLKKKLLVKW | 28.5 |

### B.2  Additional Ablations

In Table B.8 and Figure B.2, we provide additional analysis of the effect of adding two optional enhancements 1) the Literature RAG Tool (LT) and 2) Task Awareness (TA) to `PABLO`, as described in Section 3. Table B.8 provides results on all 10 selected GuacaMol tasks while Figure B.2 provides results on three representative GuacaMol tasks. As we saw in Figure 3, adding the LT leads to significant performance improvements on some tasks. In Figure B.2, notice that adding TA leads yields the largest consistent gains across tasks in both convergence speed and final objective value. In Figure B.3, we

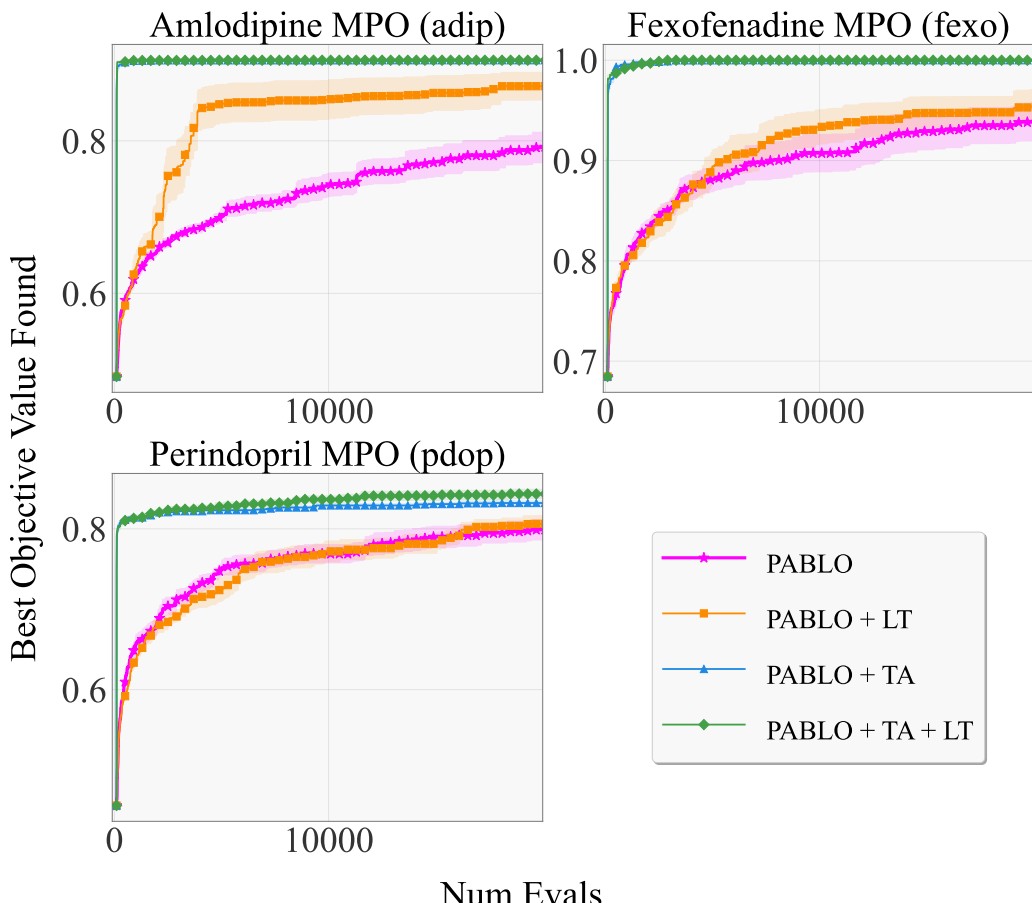

*Figure B.2.* Ablations on representative GuacaMol tasks showing the contribution of Task Awareness (TA) and the Literature RAG Tool (LT). See Table B.8 for the same comparison across all selected 10 GuacaMol tasks.

ablate the number of diverse seeds $M$ used to initialize the multi-trajectory local refinement stage of `PABLO`. We compare $M \in \{1, 2, 3, 4, 5, 6, 8, 10\}$ on three representative GuacaMol MPO tasks. The setting $M = 1$ corresponds to a simplified single-trajectory variant that always refines only the current best molecule. While this strategy can be competitive on some tasks (e.g., adip), it performs substantially worse on others (e.g., pdop and fexo), indicating that a single trajectory can become trapped in suboptimal local regions of chemical space. Using multiple diverse seeds ($M > 1$) mitigates this failure mode by enabling parallel refinement from distinct starting points, increasing the likelihood that at least one trajectory escapes poor local optima. Since it is not known *a priori* which tasks will exhibit such local traps, we adopt $M = 2$ as a robust default: it provides consistently strong performance across tasks while requiring minimal additional budget to maintain multiple trajectories. Performance is relatively stable for moderate values ($M = 2$–$6$), suggesting `PABLO` is not overly sensitive to the precise choice of $M$. In contrast, very large values ($M = 10$) can reduce performance by spreading the evaluation budget too thin across too many competing trajectories, reducing the depth of refinement per trajectory.

*Table B.8.* Ablation study comparing the effect of adding the Literature RAG Tool (LT) and Task Aware (TA) components to `PABLO` on all 10 selected GuacaMol tasks. As in Table 1, we report the mean and standard deviation of Top-1 molecules from 5 independent runs at 10K evaluations on our selected 10 tasks.

| Task | PABLO | PABLO + LT | PABLO + TA | PABLO + LT + TA |
|------|-------|------------|------------|------------------|
| med1 | 0.402±0.007 | 0.421±0.029 | 0.446±0.031 | **0.447±0.036** |
| med2 | 0.343±0.010 | 0.364±0.035 | 0.416±0.008 | **0.456±0.025** |
| pdop | 0.769±0.036 | 0.772±0.041 | 0.829±0.006 | **0.836±0.014** |
| osmb | 0.922±0.012 | 0.919±0.018 | **1.000±0.000** | 0.999±0.004 |
| adip | 0.742±0.048 | 0.853±0.080 | **0.906±0.000** | **0.906±0.000** |
| siga | 0.798±0.051 | 0.791±0.029 | **0.804±0.051** | 0.786±0.056 |
| zale | 0.600±0.040 | 0.621±0.010 | 0.716±0.002 | **0.734±0.030** |
| valt | 0.957±0.037 | 0.960±0.068 | **0.989±0.006** | 0.988±0.006 |
| rano | 0.904±0.016 | 0.904±0.012 | 0.957±0.014 | **0.961±0.021** |
| fexo | 0.910±0.059 | 0.941±0.051 | **1.000±0.000** | **1.000±0.000** |
| **Sum** | 7.347 | 7.546 | 8.063 | **8.113** |

In Figure B.4, we ablate the `MAX_FAILS` hyperparameter controlling the persistence mechanism in `PABLO`. Recall that `MAX_FAILS` specifies the number of consecutive unsuccessful attempts allowed before an agent stops persisting along its current strategy (global hypothesis or local refinement prompt) and moves on. We compare `MAX_FAILS` $\in \{1, 2, 3, 4, 5, 6, 8, 10\}$ on three representative GuacaMol MPO tasks. Overall, we find that `PABLO` is robust to the choice of `MAX_FAILS` within a reasonable range. Values in the range 2–4 achieve very similar performance across all tasks, with `MAX_FAILS`=3 performing best overall, justifying our default choice. In contrast, setting `MAX_FAILS`=1 tends to underperform, consistent with the intuition that allowing only a single attempt is overly reactive and can prematurely abandon promising strategies before they yield improvements. At the other extreme, very large values (e.g., `MAX_FAILS`=10) degrade performance by over-committing evaluations to unproductive strategies, delaying necessary exploration or prompt adaptation. These results support the use of a small but non-trivial persistence window, where agents are given multiple opportunities to exploit a promising direction while still avoiding inefficient stagnation.

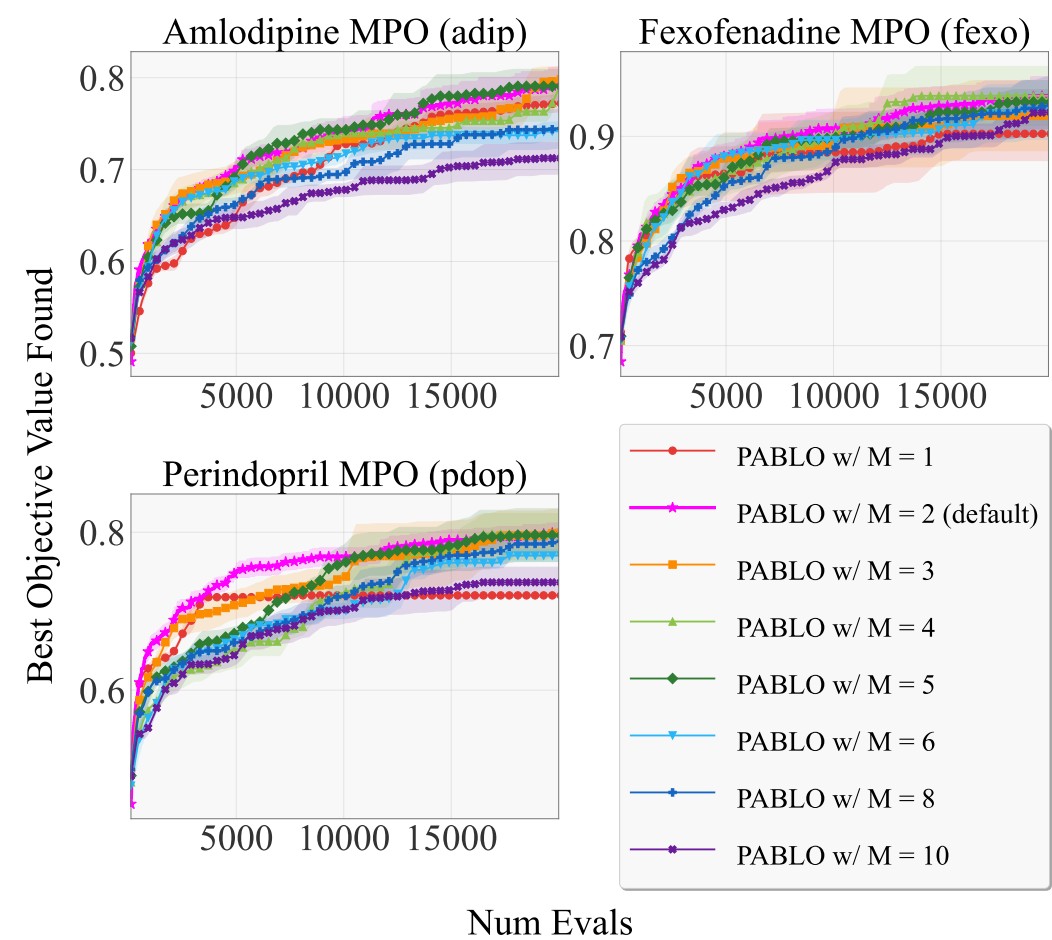

*Figure B.3.* Ablation of the number of diverse seeds ($M$) used for multi-trajectory local optimization in `PABLO` on three representative GuacaMol MPO tasks. We plot the objective value of the best molecule found so far versus the number of black-box evaluations.

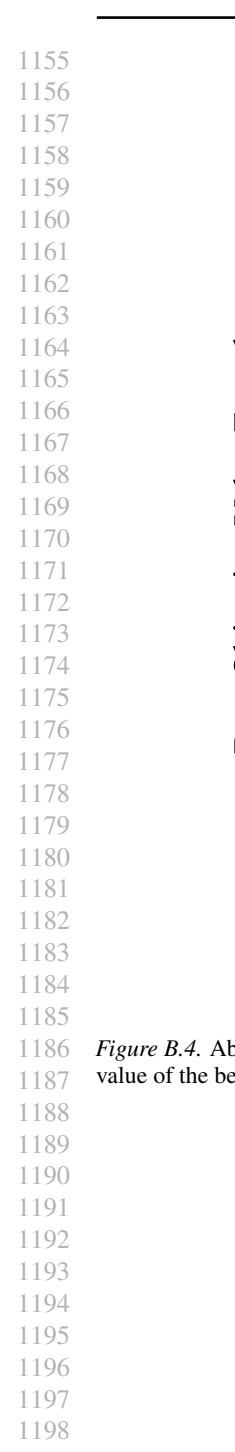

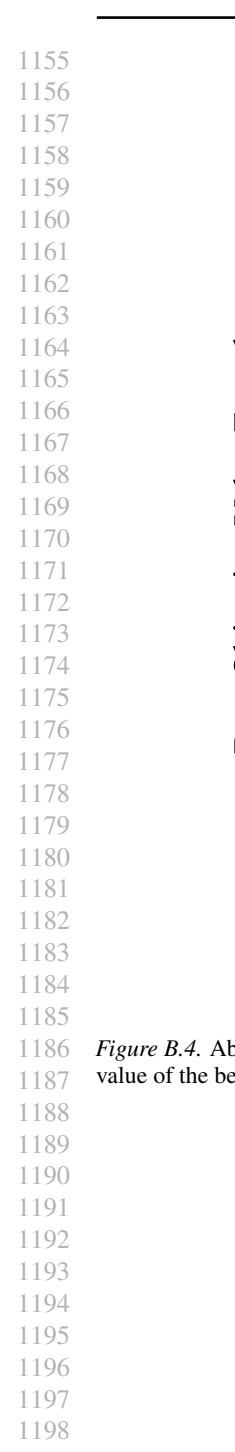

*Figure B.4.* Ablation of the `MAX_FAILS` hyperparameter in `PABLO` on three representative GuacaMol MPO tasks. We plot the objective value of the best molecule found so far versus the number of black-box evaluations.

## C  Limitations of `PABLO`

`PABLO` leverages LLMs as optimization agents, which introduces practical limitations related to (i) inference cost and (ii) agent latency/runtime.

**LLM inference cost.**   A primary limitation of `PABLO` is that it requires LLM queries throughout optimization. To quantify this overhead, we report token usage statistics in Figure C.1 over $n = 100$ full optimization runs of `PABLO`. As shown in the figure, each run consumes on average 2.50M tokens on Intern S1 (0.65M input; 1.84M output) and 16.71M tokens on GPT-OSS-120B (2.00M input; 14.71M output), for a total of 19.21M tokens per optimization run. Figure C.1 also shows that most tokens are generated by the Worker Agent model (GPT-OSS-120B output tokens), reflecting `PABLO`'s deliberate design choice to allocate the bulk of generation to a fast, relatively inexpensive model.

To provide a concrete estimate of what this would cost under a hosted API deployment, we compute the *equivalent inference cost per run* using public token pricing for GPT-OSS-120B ($0.09/M input tokens and $0.45/M output tokens). Under these rates, the mean GPT-OSS-120B cost per run is:

$$\text{Cost}_{\text{OSS}} = (2.00 \times 0.09) + (14.71 \times 0.45) \approx \$6.80.$$

Intern S1 pricing depends on the deployment/provider. Under a representative hosted inference rate of $0.15/M input tokens and $0.60/M output tokens, the mean Intern S1 cost is:

$$\text{Cost}_{\text{IS1}} = (0.65 \times 0.15) + (1.84 \times 0.60) \approx \$1.20.$$

This yields an estimated total hosted inference cost of:

$$\text{Cost}_{\text{Total}} \approx \$8.00 \ \text{ per optimization run}.$$

Importantly, in our experiments we did *not* incur per-token API costs, since we self-hosted both LLMs on our institutional GPU compute cluster. Intern S1 was served using `vLLM` on 4×B200 GPUs, and GPT-OSS-120B was served using `SGLang` on 2×B200 GPUs (Kwon et al., 2023; Zheng et al., 2024). We report the hosted-cost estimate above to make the computational overhead transparent and comparable for researchers deploying `PABLO` through public APIs.

While non-trivial, this inference cost is typically small relative to downstream experimental validation. In wet-lab design pipelines, synthesizing and assaying even a small number of candidates can cost orders of magnitude more than LLM inference. Moreover, `PABLO` is cost-aware by design: the majority of candidate generation is performed by the cheaper Worker model (GPT-OSS-120B), while the higher-capacity model (Intern S1) is used strategically for global reasoning (Explorer Agent) and prompt/task synthesis (Planner Agent), thereby limiting expensive calls.

**Runtime and agent latency.**   A second limitation is wall-clock runtime, since optimization proceeds through sequential agent calls and oracle evaluations. In our experiments, runs of `PABLO` required an average of 99.7 hours to reach 20K total black-box evaluations. Although agent latency can be a bottleneck, this runtime is on par with (and often faster than) strong baselines that involve expensive training, model refitting, or repeated sampling, such as NF-BO, LLAMBO, and related deep generative optimization methods. In practice, `PABLO` benefits from the high throughput of GPT-OSS-120B, which accounts for most generation tokens, enabling competitive end-to-end runtime despite the hierarchical agentic loop.

Overall, `PABLO` trades increased inference overhead for substantially improved sample efficiency and solution quality, which is often the dominant priority in biological design settings where oracle evaluations correspond to costly experiments.

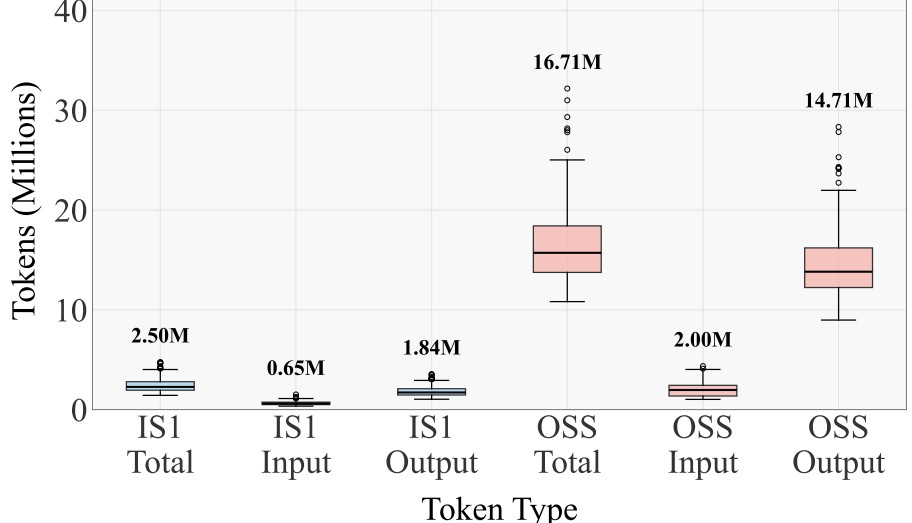

*Figure C.1.* Token usage distribution across optimization runs for the two LLMs used in `PABLO`: Intern S1 (IS1) and GPT-OSS-120B (OSS). We report input tokens, output tokens, and total tokens per run, where all runs use a budget of 20K total black-box function evaluations.

## D   Additional Implementation Details

In this section, we provide additional implementation details for `PABLO`.

### D.1   `PABLO` Hyperparameters

All hyperparameters were fixed for all runs of `PABLO` across tasks unless otherwise noted.

**Global context construction.**   The global context $C_{\text{global}}$ provided to the Explorer Agent has a fixed size of 20 candidates in all experiments. We include the top-$k$=8 best-performing candidates as positive exemplars. The remaining 12 candidates are sampled from the rest of the optimization history at approximately uniform rank intervals (with a random offset), to ensure that the context spans the observed performance spectrum.

**Success persistence.**   The persistence loop in both global and local phases is controlled by a patience counter `MAX_FAILS`. In all experiments in Section 3, we use `MAX_FAILS` = 3. An ablation study justifying this default is shown in Figure B.4.

**Number of diverse seeds.**   For multi-trajectory local optimization, we use $M = 2$ diverse seeds in all experiments. An ablation over $M$ is provided in Figure B.3, which shows that $M = 2$ provides a strong trade-off between robustness to local optima and efficient allocation of evaluation budget.

**Distance function and diversity threshold for $S_{\text{seed}}$.**   As discussed in Section 2, selection of $S_{\text{seed}} = \{x^{(1)}, \ldots, x^{(M)}\}$ on each round of `PABLO` involves greedy selection under a diversity threshold, using a domain-specific distance function $\text{dist}(\cdot, \cdot)$. For all molecule design tasks in this paper, we used fingerprint similarity (FPS) to measure distance between molecules (computed using RDKit) and a threshold of $0.5$. Thus, we require that ever pair of molecules in $S_{\text{seed}}$ has FPS no higher than $0.5$. For all peptide design tasks, we used normalized Levenshtein edit distance to measure distance between peptide amino acid sequences. In particular, normalized edit distance between two peptide sequences means we compute Levenshtein edit distance between the two sequences, and then divide by the length of the shorter sequence. For all peptide design runs we use a diversity threshold of $0.75$. Thus we require that ever pair of peptides in $S_{\text{seed}}$ has normalized edit distance of at least $0.75$.

**Hyperparameters for diverse portfolio optimization.**   For runs of our method to optimize a diverse set of $M = 20$ peptides (rather than just one peptide), we similarity require a means of measuring distance between peptides and a diversity threshold (the minimum distance we want to require between peptide sequences in our diverse portfolio). For this, we use the same distance function and diversity threshold as we do for constructing the diverse set of peptides $S_{\text{seed}}$. In particular,

we measure distance between two peptide sequences using normalized edit distance: (Levenshtein edit distance) / (the length of the shorter sequence), and we use the same diversity threshold of threshold 0.75. Thus, we require all pairs of peptides in the optimized diverse portfolio to have a normalized Levenshtein edit distance of at least 0.75.

**Task Registry initialization and size.**   The Task Registry is initialized with 3 domain-specific default tasks. These default tasks are hand-designed to illustrate simple, standard modification strategies for the given domain. The exact default tasks used for each domain (molecules and peptides) are listed in Section I.1.

The Task Registry has a fixed maximum capacity of MAX_TASKS=20 tasks. When adding a new task would exceed this limit, the worst-performing non-default task (based on empirical success rate) is pruned. We chose a capacity of 20 to maintain a diverse library of local search strategies while avoiding excessive context length that could degrade Planner Agent reasoning performance.

**Planner Agent task selection** ($|\mathcal{P}_{\mathbf{work}}|$).   At each Planner Agent invocation, the prompt instructs the agent to return approximately 8–10 task prompts, consisting of a mixture of existing and newly synthesized tasks. The exact number is determined by the agent at generation time.

**Explorer Agent batch size** ($|B_{\mathbf{global}}|$).   At each global search step, the Explorer Agent is prompted to generate 10–20 new candidates. The precise number of outputs is determined by the agent.

**Worker batch size** ($|B_{\mathbf{loc}}|$).   During local refinement, each Worker Agent is prompted to generate 5–10 candidate modifications per iteration. The exact number is determined by the agent at generation time.

### D.2   Initialization Data

In all experiments, PABLO was initialized with 100 data points. For AMP design tasks, we initialize with the 10 template sequences in Table E.3 plus 90 additional sequences generated via random mutations to those templates. For molecular design tasks, following Lee et al. (2025), we initialize with the first 100 SMILES from the GuacaMol dataset (Brown et al., 2019).

All initialization points are counted toward the total evaluation budget of 20,000 oracle calls and are included on the x-axis in all plots in Section 3. For baseline methods, following Gao et al. (2022a), we adopt the initialization strategy specified in each baseline's original implementation when available; otherwise, we use the same 100 initialization data points as for PABLO to ensure fair comparison.

**Initialization under zero-signal regimes.**   All GuacaMol tasks have objective functions bounded in $[0, 1]$. This can sometimes produce large regions of search space with identical minimum score (exactly 0.0). In such cases, an initial dataset may contain no score variation, providing no signal about how candidate structure influences the objective. To avoid this degenerate regime, we ensure that the initialization set contains at least one molecule with non-zero score before beginning optimization.

Concretely, if all 100 initial molecules have score 0.0, we continue sampling additional SMILES uniformly at random from the GuacaMol dataset and evaluating them until at least one molecule with score $> 0$ is found. These additional initialization evaluations are still counted toward the total budget and are included on the x-axis of all plots.

This procedure was only required for the Valsartan SMARTS task. In our runs, between approximately 200 and 1400 additional molecules were evaluated before observing a non-zero score. The highest score observed during this random initialization phase was extremely small ($4.87 \times 10^{-36}$), but even this minimal deviation from zero was sufficient to provide a usable optimization signal. With a single non-zero example, PABLO was then able to reliably discover molecules with near-optimal scores $> 0.9$. In contrast, when initialized with only zero-scoring molecules, PABLO made no progress and remained at 0.0 for the entire optimization run.

We therefore recommend this strategy for benchmarks with hard lower bounds and large zero-signal regions, where initial samples may all share the same minimum score. In such cases, allocating additional budget to random exploration until at least some score variation is observed can dramatically improve optimization performance.

This issue does not arise in our peptide experiments or in many realistic experimental or predictive oracles, where scores are continuous and do not exhibit large flat regions at an absolute minimum. In those settings, even small variations in initialization data provide sufficient signal for PABLO to begin effective optimization without any special initialization procedure.

*(a)* Query molecule

*(b)* Retrieved: **9a** (PMID: 2342074)

*"Thus, the 1,2,3-triazole derivatives with a 5-carbethoxy (9a) and 5-carboxamido (11a) substituent are some 5-fold less active as calcium antagonists than their corresponding 4-isomers (8a and 10a, respectively)."*

*Figure D.1.* Molecule RAG retrieval example. Given a query structure, we retrieve papers discussing structurally similar molecules from PubMed.

# E    Antimicrobial Peptide (AMP) Optimization Task Details

*Table E.1.* Names of the 11 target bacteria used for the AMP design task.

| Objective ID | Target Pathogenic Bacteria |
|---|---|
| B1 | A. baumannii ATCC 19606 |
| B2 | E. coli ATCC 11775 |
| B3 | E. coli AIC221 |
| B4 | E. coli AIC222-CRE |
| B5 | K. pneumoniae ATCC 13883 |
| B6 | P. aeruginosa PAO1 |
| B7 | P. aeruginosa PA14 |
| B8 | S. aureus ATCC 12600 |
| B9 | S. aureus ATCC BAA-1556-MRSA |
| B10 | E. faecalis ATCC 700802-VRE |
| B11 | E. faecium ATCC 700221-VRE |

*Table E.2.* Names of the 9 extra (non-target) bacteria use for *in vitro* experiments testing broad spectrum activity of optimized peptides and included in Figure B.1.

| Objective ID | Pathogenic Bacteria |
|---|---|
| B12 | A. baumannii ATCC BAA-1605-CGTPACCIMRA |
| B13 | E. cloacae ATCC 13047 |
| B14 | E. coli ATCC BAA-3170-CRE |
| B15 | E. coli K-12 BW25113 |
| B16 | K. pneumoniae ATCC BAA-2342-EIRK |
| B17 | P. aeruginosa ATCC BAA-3197-FBCRP |
| B18 | S. enterica ATCC 9150 |
| B19 | S. enterica Typhimurtium ATCC 700720 |
| B20 | B. subtilis ATCC 23857 |

Table E.1 specifies the 11 target bacteria used for the AMP design task from Section 3. Specifically, we designed peptides to optimize the mean predicted MIC across the 11 target bacteria in Table E.1, where in-silico MICs were predicted using the Apex 1.1 model as an oracle (Torres et al., 2025). The first seven bacteria are Gram negative bacteria (Objective IDs B1-B7) and the last four (Objective IDs B8-B11) are Gram positive.

Table E.3 gives the 10 template amino acid sequences used for the "template constrained" variation of the AMP design task from Section 3.

*Table E.3.* Template amino acid sequences used for the "template constrained" AMP design task.

| Template Amino Acid Sequences |
|:---:|
| RACLHARSIARLHKRWRPVHQGLGLK |
| KTLKIIRLLF |
| KRKRGLKLATALSLNNKF |
| KIYKKLSTPPFTLNIRTLPKVKFPK |
| RMARNLVRYVQGLKKKKVI |
| RNLVRYVQGLKKKKVIVIPVGIGPHANIK |
| CVLLFSQLPAVKARGTKHRIKWNRK |
| GHLLIHLIGKATLAL |
| RQKNHGIHFRVLAKALR |
| HWITINTIKLSISLKI |

# F  *In Vitro* Minimal Inhibitory Concentration (MIC) Experimental Methods

**Peptide synthesis and characterization.**    Peptides were synthesized on an automated peptide synthesizer (Symphony X, Gyros Protein Technologies) by standard 9-fluorenylmethyloxycarbonyl (Fmoc)-based solid-phase peptide synthesis (SPPS) on Fmoc-protected amino acid-Wang resins (100–200 mesh). In addition to preloaded resins, standard Fmoc-protected amino acids were employed for chain elongation. N,N-Dimethylformamide (DMF) was used as the primary solvent throughout synthesis. Stock solutions included: $500 \, \mathrm{mmol \, L^{-1}}$ Fmoc-protected amino acids in DMF, a coupling mixture of HBTU ($450 \, \mathrm{mmol \, L^{-1}}$) and N-methylmorpholine (NMM, $900 \, \mathrm{mmol \, L^{-1}}$) in DMF, and 20% (v/v) piperidine in DMF for Fmoc deprotection. After synthesis, peptides were deprotected and cleaved from the resin using a cleavage cocktail of trifluoroacetic (TFA)/triisopropylsilane (TIS)/dithiothreitol (DTT)/water (92.8% v/v, 1.1% v/v, 0.9% w/v, 4.8%, w/w) for 2.5 hours with stirring at room temperature. The resin was removed by vacuum filtration, and the peptide-containing solution was collected. Crude peptides were precipitated with cold diethyl ether and incubated for 20 min at $-20\,^{\circ}\mathrm{C}$, pelleted by centrifugation, and washed once more with cold diethyl ether. The resulting pellets were dissolved in 0.1% (v/v) aqueous formic acid and incubated overnight at $-20\,^{\circ}\mathrm{C}$, followed by lyophilization to obtain dried peptides. For characterization, peptides were dried, reconstituted in 0.1% formic acid, and quantified spectrophotometrically. Peptide separations were performed on a Waters XBridge $C_{18}$ column ($4.6 \times 50 \, \mathrm{mm}$, $3.5 \, \mathrm{\mu m}$, $120 \, \text{Å}$) at room temperature using a conventional high-performance liquid chromatography (HPLC) system. Mobile phases were water with 0.1% formic acid (solvent A) and acetonitrile with 0.1% formic acid (solvent B). A linear gradient of 1–95% B over 7 min was applied at $1.5 \mathrm{mL \, min^{-1}}$. UV detection was monitored at 220nm. Eluates were analyzed on Waters SQ Detector 2 with electrospray ionization in positive mode. Full scan spectra were collected over m/z 100–2,000. Selected Ion Recording (SIR) was used for targeted peptides. Source conditions were capillary voltage 3.0kV, cone voltage 25-40 V, source temperature $120\,^{\circ}\mathrm{C}$, and desolvation temperature $350\,^{\circ}\mathrm{C}$. Mass spectra were processed with MassLynx software. Observed peptide masses were compared with theoretical values, and quantitative analysis was based on integrated SIR peak areas.

**Bacterial Strains and Growth Conditions.**    The bacterial panel utilized in this study consisted of the following pathogenic strains: *Acinetobacter baumannii* ATCC 19606; *A. baumannii* ATCC BAA-1605 (resistant to ceftazidime, gentamicin, ticarcillin, piperacillin, aztreonam, cefepime, ciprofloxacin, imipenem, and meropenem); *Escherichia coli* ATCC 11775; *E. coli* AIC221 [MG1655 phnE_2::FRT, polymyxin-sensitive control]; *E. coli* AIC222 [MG1655 pmrA53 phnE_2::FRT, polymyxin-resistant]; *E. coli* ATCC BAA-3170 (resistant to colistin and polymyxin B); *E. coli* K-12 BW25113; *Enterobacter cloacae* ATCC 13047; *Klebsiella pneumoniae* ATCC 13883; *K. pneumoniae* ATCC BAA-2342 (resistant to ertapenem and imipenem); *Pseudomonas aeruginosa* PAO1; *P. aeruginosa* PA14; *P. aeruginosa* ATCC BAA-3197 (resistant to fluoroquinolones, $\beta$-lactams, and carbapenems); *Salmonella enterica* ATCC 9150; *S. enterica* subsp. *enterica* Typhimurium ATCC 700720; *Bacillus subtilis* ATCC 23857; *Staphylococcus aureus* ATCC 12600; *S. aureus* ATCC BAA-1556 (methicillin-resistant); *Enterococcus faecalis* ATCC 700802 (vancomycin-resistant); and *Enterococcus faecium* ATCC 700221 (vancomycin-resistant). *P. aeruginosa* strains were propagated on Pseudomonas Isolation Agar, whereas all other species were maintained on Luria-Bertani (LB) agar and broth. For each assay, cultures were initiated from single colonies, incubated overnight at $37\,^{\circ}\mathrm{C}$, and subsequently diluted 1:100 into fresh medium to obtain cells in mid-logarithmic phase.

**Minimal Inhibitory Concentration (MIC) Determination.**    MIC values were established using the standard broth microdilution method in untreated 96-well plates. Test peptides were dissolved in sterile water and prepared as twofold serial dilutions ranging from 1 to $64 \, \mathrm{\mu mol \, L^{-1}}$. Each dilution was combined at a 1:1 ratio with LB broth containing $4 \times 10^6$ CFU $\mathrm{mL^{-1}}$ of the target bacterial strain. Plates were incubated at $37\,^{\circ}\mathrm{C}$ for 24 hours, and the MIC was the lowest peptide concentration that inhibited visible bacterial growth. All experiments were conducted independently in triplicate.

## G  Example Global Context ($C_{\text{global}}$)

The global context $C_{\text{global}}$ provided to the Explorer Agent and Planner Agent consists of scored candidates drawn from the optimization history using the coverage sampling procedure described in Section 2.1. The exact contents of $C_{\text{global}}$ change throughout optimization as new data are collected. Below we show one representative snapshot of $C_{\text{global}}$ from each domain.

$C_{\text{global}}$ **Snapshop for Molecules (score, SMILES):**

```
0.4833:   CC(C)=CCn1c2cc(=O)ccc-2nc2c(C(N)=O)cccc21
0.4695:   CC1(C)Oc2ccc(C#N)cc2C(NC(=O)c2ccccn2)C1O
0.4447:   CCOC(=O)C1C(=O)NC(c2ccccc2)=NC1c1cccc(Br)c1
0.4427:   CN1CC(N(Cc2ccc(F)cc2F)c2ccc(C#N)c(Cl)c2)CC1=O
0.4426:   CN1C(=O)C(c2ccc(C#Cc3ccnc(Cl)c3)cn2)CC1(C)C
0.4389:   CC(O)(COc1ccccc1)C(=O)N1CCc2c(C#N)cccc21
0.4294:   Cc1ccc(-n2ncc3c(=O)n(-c4cccc(C)c4)c(=O)[nH]c32)cc1
0.4171:   CC(=O)N1CCCC12c1ccccc1-c1nc(O)c3nccn3c12
0.2909:   CC1Cn2c(nnc2-c2ccccn2)CN1C(=O)c1cccc(Cl)c1Cl
0.1400:   O=c1c(C2=Nc3ccccc3S(=O(=O)N2)c(O)c2cc(F)ccc2n1CCC1CC1
0.07337:  O=C1OC(CO)(COC(=O)c2ccccc2)CC1=Cc1ccc([N+](=O)[O-])cc1
0.03613:  COc1ccc(S(=O)(=O)NC(C)C(=O)Nc2ccc3c(c2)OCO3)cc1OC
0.01600:  O=c1ccncn1CC1(O)CC2NCCCC2O1
0.005560:  CC1CC2CCC3OC(=O)C(C1SCCS(=O)(=O)O)C23O
0.001304:  Nc1ncnc2c1c(-c1cc3ccccc3c3ccccc13)cn2C1OC(CO)C(O)C1O
1.957e-04:  CC(=CC(=O)O)C(O)P(=O)(O)CCC(N)C(=O)O
1.044e-05:  COc1ccc2c(c1)N=C(N)NC2
1.205e-07:  COc1cccc2c1CCC1CN(CCn3c(=O)[nH]c4c(OC)c(OC)ccc4c3=O)CC21
1.657e-11:  NC(=O)c1cc(CI)[nH]n1
3.136e-24:  CSCCC(NC(=O)OC(C)(C)C)C(=O)N1CCC(C(=O)NC(C)C(=O)NC(C)c2ccccc2)CC1
```

$C_{\text{global}}$ **Snapshop for Peptides (MIC, sequence):**

```
85.12:   KLLKIRRLWF
90.12:   WRKRGLKLATWLSLLNKF
95.11:   KWLKIIRLLF
99.25:   KHLKIMRLLW
106.4:   LGLKIIRLLF
109.4:   MGTKPMIKVRRKRLKQFVAK
111.3:   LGIRVRWMKTYTYCKKIKMFK
114.5:   KRVRINRHKRVLKKPVNDFPYLQF
270.0:   KNLKIIRLLA
313.2:   KHYKKLSTPPFTLNIRTLPKVKFPK
345.6:   KRKRGLKLGTQLSLNNKF
373.6:   RACLHARSIWRLHLRWRPVHQGLKLK
400.3:   RQKNHGIHFRVKANALR
432.5:   RQKNHGIRDRVLAKALR
466.2:   YAFWTFVPHPVVRFINRIP
486.6:   ANHLFTFAFPPCKILQNGRNQH
497.7:   QNQQKGATPGEFYKQFIQHC
503.5:   NSLYALGEYQLTKRY
507.3:   DNCQQVYYAWSHHHPMSSDLPA
510.3:   ESGNCQFDNEFEEHN
```

## H  Example Explorer Agent Prompts

The Explorer Agent is prompted using (i) a snapshot of the current global context $C_{\text{global}}$ (inserted into the prompt as a score-sorted list of previously evaluated candidates) and (ii) the current best observed score, which is used to set an explicit improvement target. Below we provide representative prompts used in each domain. See Section G for examples of the $C_{\text{global}}$ that is included in each full example prompt below.

**Explorer Example Prompt for Molecules:**

```
You are an expert molecule designer.  Generate molecules that you reason are
most likely to score HIGHER than any in the provided MOLECULE-SCORE DATA.
## MOLECULE-SCORE DATA (sorted high to low)
{C_global}
## TASK
Think step-by-step:
1.  **Analyze the MOLECULE-SCORE DATA:** What molecular features correlate
with high scores?  Form 2-3 hypotheses about what the scoring function
rewards.
2.  **Generate:** Propose 10-20 NEW molecules that:
- Push your hypotheses to their LOGICAL EXTREME for maximum scores
- Combine best features from multiple top scorers
- Explore creative new structural ideas
## OUTPUT FORMAT
Return ONLY a JSON object with a list of VALID SMILES strings called
'candidates'.
Example:
{ "candidates":  ["SMILES_1", "SMILES_2", "SMILES_3", ...]  }
## GOAL
Propose new SMILES strings that could BEAT the current best score
({best_score})
```

**Explorer Example Prompt for Peptides:**

```
You are an expert antimicrobial peptide (AMP) designer.  Generate peptides
that you reason are most likely to achieve LOWER MIC than any in the provided
data.
The goal is to MINIMIZE the Minimum Inhibitory Concentration (MIC)
averaged across 11 bacterial pathogens (including E. coli, P. aeruginosa,
S. aureus/MRSA, K. pneumoniae, A. baumannii, and vancomycin-resistant
Enterococci).
LOWER MIC = BETTER (less peptide needed to inhibit bacterial growth).
## PEPTIDE-MIC DATA (sorted best to worst, LOWEST MIC first)
{C_global}
## TASK
Think step-by-step:
1.  **Analyze the PEPTIDE-MIC DATA:** What sequence features correlate with
low MIC? Form 2-3 hypotheses about what makes an effective AMP (e.g., cationic
charge, amphipathicity, hydrophobic content, length, specific motifs).
2.  **Generate:** Propose 10-20 NEW antimicrobial peptides that:
- Push your hypotheses to their LOGICAL EXTREME for minimum MIC
- Combine best features from multiple top performers
- Explore creative new sequence ideas using what you know about AMPs
## OUTPUT FORMAT
Return ONLY a JSON object with a list of VALID peptide sequences called
```

```
'candidates'.
Example:
{ "candidates": ["RKKLWLLRK", "FLPLIGKLLK", "KWKLFKKIGAVLKVL", ...]  }
## GOAL
Propose new antimicrobial peptide sequences that could BEAT the current best
MIC ({best_score})
```

## I  Example Planner Agent Prompts

The Planner Agent is responsible for generating and selecting local-search task prompts to populate the Task Registry. Each Planner Agent call is provided with: (i) a snapshot of the current global context $C_{\text{global}}$, (ii) summary performance statistics for each task in the registry, and (iii) a compact summary of the currently-available tasks.

### I.1  Task Registry Initialization

We initialize the Task Registry with three domain-specific "default" tasks. These tasks serve two purposes: (1) they represent reasonable, simple modification strategies for the given domain, and (2) they provide examples that help the Planner Agent learn the structure and style of effective task prompts.

Below we list the three default tasks used for each domain.

**Molecules - Default Task 1 (SIMILAR):**

```
TASK: Generate SMILES that are structurally similar to the input molecule.
HINTS:
1.  Modify side chains, linkers, or substituents.
2.  Keep the core scaffold mostly intact.
```

**Molecules - Default Task 2 (EXPLORE):**

```
TASK: Generate SMILES with different meaningful structural changes to the
input.
HINTS:
1.  Each output should be a distinct modification type (ring size, linker
swap, substituent move).
2.  Make significant moves, not minor tweaks.
3.  Explore broadly around the input.
```

**Molecules - Default Task 3: (SCAFFOLD_HOP):**

```
TASK: Generate scaffold hopping variations of the input molecule.
HINTS:
1.  Make large topology-level changes (new ring systems, fusion patterns).
2.  Avoid small local edits; make substantial core changes.
3.  Try:  fused<->bridged<->spiro, cyclic<->polycyclic, aromatic<->non-aromatic
cores.
```

**Peptides - Default Task 1 (SIMILAR):**

```
TASK: Generate peptides that are conservative variants of the input.
HINTS:
1.  Use similar amino acid substitutions (L<->I, D<->E, K<->R, F<->Y).
2.  Preserve overall charge and hydrophobicity patterns.
3.  Keep modifications minimal (1{2 changes).
```

**Peptides - Default Task 2 (EXPLORE):**

```
TASK: Generate peptides with meaningfully different modifications to the
input.
HINTS: 1.  Try substitutions from different amino acid classes
(polar<->hydrophobic, charged<->neutral).
2.  Vary the length by adding or removing 1{3 residues.
3.  Each output should explore a different modification strategy.
```

**Peptides - Default Task 3: (SHUFFLE):**

```
TASK: Generate peptides by rearranging amino acids in the input.
HINTS:
1.  Try swapping positions of residues.
2.  Try reversing short segments (3{5 residues).
3.  Try circular permutations (move N-terminal residues to C-terminus).
```

### I.2   Task Performance Statistics (performance_stats).

At initialization, the Planner Agent receives performance_stats = `"No performance data yet."`. Since the Task Registry is updated online, in subsequent calls performance_stats is a table of task success rates computed from the registry, where success rate is defined as the fraction of task executions that produced at least one improving candidate. The Task Registry holds at most 20 tasks at a time. The performance_stats are presented to the Planner Agent using the unique task name for each task in the current Task Registry, and it's current success rate, presented in order from highest to lowest number of attempts so far. Example snapshots of the performance_stats text included in the Planner Agent prompt for each domain are provided below:

**Example performance_stats Text - Molecules:**

```
SIMILAR: 15/42 (36%)
CROSSOVER: 9/28 (32%)
POLARITY_MOD: 7/16 (44%)
EXPLORE: 6/14 (43%)
SCAFFOLD_HOP: 5/12 (42%)
ELECTROSTATIC_OPTIMIZATION: 4/10 (40%)
SWAP_FUNCTIONAL_GROUPS: 3/9 (33%)
SIMPLIFY: 2/8 (25%)
RING_EXPANSION: 0/6 (0%)
```

**Example performance_stats Text - Peptides:**

```
EXPLORE: 24/51 (47%)
SIMILAR: 10/45 (22%)
LENGTH_VARIATION: 8/20 (40%)
SHUFFLE: 2/13 (15%)
SWAP_ANALOG: 0/12 (0%)
SIMPLIFY: 4/8 (50%)
SWAP_ANALOG_V2 :  1/3 (33%)
```

### I.3   Existing Task Summary (existing_tasks_summary).

To avoid spending prompt budget on full task descriptions, we provide an *existing_tasks_summary* that lists each unique task name along with a truncated preview of the task description (the first 100 characters in the first line). For example, in the molecule domain this summary will contain entries such as
`EXPLORE: TASK: Generate SMILES with different meaningful structural changes...`,

SCAFFOLD_HOP: TASK: Generate scaffold hopping variations of the input molecule...,
and SIMILAR: TASK: Generate SMILES that are structurally similar....
As optimization progresses, this list will also include newly created tasks proposed by the Planner Agent (e.g., NEW_TASK1:
TASK: ...), in addition to the original default tasks. A similar structure is used for peptides (e.g., EXPLORE, SHUFFLE,
SIMILAR, along with all Planner Agent–generated tasks in the Task Registry).

### I.4   Full Planner Agent Prompt Examples.

Here we provide examples of full Planner agent prompts for each domain. See Section G for examples of the $C_{\text{global}}$ that is
included in each full example prompt below.

**Molecules - Full Planner Agent Prompt Example:**

```
You are a prompt generator for a molecular optimization system that is trying
to find the highest-scoring molecules.
## MOLECULE-SCORE DATA (sorted high to low)
{C_global}
## TASK PERFORMANCE (success rate = score improvements / attempts)
{performance_stats}
## EXISTING TASKS (you can reuse by name or create new ones)
{existing_tasks_summary}
---
## YOUR ANALYSIS PROCESS
1.  **Study the Score Gradient:** Compare molecules with SIMILAR scores.  What
small structural change caused one to score slightly higher than another?
These small differences are highly informative.
2.  **High vs Low Contrast:** What features appear in top scorers but not
low scorers?  (ring types, chain lengths, functional groups, heteroatoms,
flexibility)
3.  **Identify Gaps:** What types of modifications have NOT been tried yet?
What regions of chemical space remain unexplored?
## YOUR GOAL
Generate task prompts that help a smaller LLM:
- **EXPLOIT:** Make targeted modifications based on patterns you observe in
the score gradient
- **EXPLORE:** Try diverse, creative modifications to discover new promising
regions
We are often stuck at local optima.  To escape, we need BOTH:
- Smart exploitation of what seems to work
- Broad exploration of untried modification types
## YOUR OUTPUT FORMAT
Return a JSON object with task names as keys and task descriptions as values.
- To REUSE an existing task:  "TASK_NAME":  "USE_EXISTING"
- To CREATE a new task:  "NEW_NAME":  "TASK: ...  HINTS: ..."
Example:  {
"SIMILAR":  "USE_EXISTING",
"EXPLORE":  "USE_EXISTING",
"FUSER":  "TASK: Generate variants of the input molecule by fusing adjacent
rings into polycyclic cores.
HINTS:
1.  Merge rings connected by short chains (1-2 atoms).
2.  Create bicyclic or tricyclic fused systems.
3.  Preserve peripheral substituents."
}
## GUIDELINES
```

```
1.  Output **8-10 tasks total** – a mix of existing and new.
2.  Include 2-3 EXPLOITATION tasks (targeted at patterns you observed).
3.  Include 2-3 EXPLORATION tasks (creative, untried modification types).
4.  Include 2-4 reliable existing tasks that have (>0%) success rates.
5.  If a task 0 successes, avoid it or create an improved version (e.g.,
TASK_NAME_V2).
6.  Keep new task descriptions concise (3-5 hints max).
7.  New task names: SHORT, DESCRIPTIVE, ALL_CAPS (e.g., ATOM_SWAP, STABILIZE,
RIGIDIFY).
## CREATIVE EXPLORATION IDEAS
Consider tasks involving:
- Specific functional groups
- Specific atoms
- Specific ring modifications (aromatic<->aliphatic, 5-ring<->6-ring, fusion,
 spiro)
- Chain modifications (extend, shorten, branch, cyclize)
- Polarity changes (add polar groups, remove polar groups)
- Symmetry changes (break symmetry, add symmetry)
- Conformational changes (rigidify, flexibilize, add rotatable bonds)
Think creatively! What modification might lead to a breakthrough?
```

**Peptides - Full Planner Agent Prompt Example:**

```
You are a prompt generator for an antimicrobial peptide (AMP) optimization
system that is trying to find peptides with the LOWEST MIC (Minimum Inhibitory
Concentration).
LOWER MIC = BETTER (less peptide needed to inhibit bacterial growth).
## PEPTIDE-MIC DATA (sorted best to worst, LOWEST MIC first)
{C_global}
## TASK PERFORMANCE (success rate = MIC improvements / attempts)
{performance_stats}
## EXISTING TASKS (you can reuse by name or create new ones)
{existing_tasks_summary}
---
## YOUR ANALYSIS PROCESS
1.  **Study the Score Gradient:** Compare peptides with SIMILAR MICs. What
small sequence change caused one to have slightly lower MIC than another?
These small differences are highly informative.
2.  **High vs Low Contrast:** What features appear in top performers but not
poor performers? (charge distribution, hydrophobic patches, length, specific
motifs)
3.  **Identify Gaps:** What types of modifications have NOT been tried yet?
What regions of sequence space remain unexplored?
## YOUR GOAL
Generate task prompts that help a smaller LLM:
- **EXPLOIT:** Make targeted modifications based on patterns you observe in
the score gradient
- **EXPLORE:** Try diverse, creative modifications to discover new promising
 regions
We are often stuck at local optima. To escape, we need BOTH:
- Smart exploitation of what seems to work
- Broad exploration of untried modification types
```

```
## YOUR OUTPUT FORMAT
Return a JSON object with task names as keys and task descriptions as values.
- To REUSE an existing task:  {"TASK_NAME":  "USE_EXISTING"}
- To CREATE a new task:  {"NEW_NAME":  "TASK: ...   HINTS: ..."}
Example:
{
"SIMILAR":   "USE_EXISTING",
"EXPLORE":   "USE_EXISTING",
"CHARGE_BOOST":   "TASK: Increase the net positive charge of the input peptide.
HINTS:
1.   Replace neutral residues with K or R.
2.   Replace acidic residues (D, E) with neutral or basic ones.
3.   Add K or R at termini."
}
## GUIDELINES
1.   Output **8-10 tasks total** - a mix of existing and new.
2.   Include 2-3 EXPLOITATION tasks (targeted at patterns you observed).
3.   Include 2-3 EXPLORATION tasks (creative, untried modification types).
4.   Include 2-4 reliable existing tasks that have (>0%) success rates.
5.   If a task 0 successes, avoid it or create an improved version (e.g.,
TASK_NAME_V2).
6.   Keep new task descriptions concise (3-5 hints max).
7.   New task names:  SHORT, DESCRIPTIVE, ALL_CAPS (e.g., CHARGE_BOOST,
HELIX_FORM, TRUNCATE).
## CREATIVE EXPLORATION IDEAS FOR AMPs
Consider tasks involving:
- Charge modifications (increase/decrease cationic character)
- Hydrophobicity changes (add/remove hydrophobic residues)
- Secondary structure (promote helix, add proline kinks)
- Length modifications (truncate, extend, repeat motifs)
- Amphipathicity (arrange polar/nonpolar faces)
- Specific motif insertions (WRW, KWK, etc.)
Think creatively!   What modification might lead to a breakthrough?
```

## J   Example Worker-Agent Prompts

Unlike the Explorer and Planner Agents, which use a single monolithic prompt that is dynamically updated with the current global context, Worker Agents use a two-part prompting structure consisting of a **system prompt** and a **generation-time prompt**.

### J.1   Worker system prompt.

For a given task prompt $p \in \mathcal{P}_{\text{work}}$ generated by the Planner Agent, the Worker system prompt is constructed as:

$$\texttt{SystemPrompt} = \texttt{prefix}_{\text{domain}} + p + \texttt{suffix}_{\text{domain}}.$$

The prefix and suffix are fixed (unchanging) for a given domain, while the task description $p$ varies. Since examples of $p$ output by the Planner Agent are provided in Section I.1 and Section K, here we provide the domain-specific prefix and suffix needed to complete the Worker Agent system prompt for each domain.

**Molecule domain prefix:**

```
You are an expert molecule generator operating in SMILES space.
INPUT: You will be given a single input molecule in the prompt.
```

**Molecule domain suffix (fixed):**

```
OUTPUT FORMAT (REQUIRED): Return ONLY a JSON object with a list of 5-10 SMILES
strings called 'candidates'.
```

**Peptide domain prefix (fixed):**

```
You are an expert peptide generator operating in amino acid sequence space.
INPUT: You will be given a single input peptide in the prompt.
```

**Peptide domain suffix (fixed):**

```
OUTPUT FORMAT (REQUIRED): Return ONLY a JSON object with a list of 5-10
peptide sequences called 'candidates'.
```

Note that the prefix and suffix text for different domains is nearly identical, with only small changes needed to specify what type of biological objects we want the Worker Agent to generate (e.g., SMILES strings vs. peptide sequences).

**Example full Worker system prompts.** We additionally provide an example of a full worker system prompt for each domain, using one example default task prompt $p$ from each domain to construct the full system prompts as:

$$\text{SystemPrompt} = \text{prefix}_{\text{domain}} + p + \text{suffix}_{\text{domain}}.$$

.

**Full Worker System Prompt Example - Molecules:**

```
You are an expert molecule generator operating in SMILES space.
INPUT: You will be given a single input molecule in the prompt.
TASK: Generate SMILES that are structurally similar to the input molecule.
HINTS:
1.  Modify side chains, linkers, or substituents.
2.  Keep the core scaffold mostly intact.
OUTPUT FORMAT (REQUIRED): Return ONLY a JSON object with a list of 5-10 SMILES
strings called 'candidates'.
```

**Full Worker System Prompt Example - Peptides:**

```
You are an expert peptide generator operating in amino acid sequence space.
INPUT: You will be given a single input peptide in the prompt.
TASK: Generate peptides by rearranging amino acids in the input.
HINTS:
1.  Try swapping positions of residues.
2.  Try reversing short segments (3{5 residues).
3.  Try circular permutations (move N-terminal residues to C-terminus).
OUTPUT FORMAT (REQUIRED): Return ONLY a JSON object with a list of 5-10
peptide sequences called 'candidates'.
```

### J.2 Worker Generation-time Prompt.

While the system prompt encodes the task description, the specific seed sequence or molecule is provided separately at generation time. For a current seed $x_{\text{curr}}$ (see Algorithm 1), the generation-time prompt is:

**Worker Agent Generate-time Prompt - Molecules:**

```
Input Molecule:  {x_curr} Modify it to generate 5-10 VALID SMILES strings.
```

**Worker Agent Generate-time Prompt - Peptides:**

```
Input Peptide:  {x_curr} Modify it to generate 5-10 new peptides.
```

This separation allows the same Worker Agent (same system prompt) to be reused across different seeds during the local search persistence loop. A new Worker Agent (with a new system prompt) is instantiated only when switching to a different task prompt $p \in \mathcal{P}_{\text{work}}$, where $\mathcal{P}_{\text{work}}$ is the latest set of task prompts generated by the Planner Agent.

# K  Example Tasks Generated by the Planner Agent

Throughout optimization, the Planner Agent generates new tasks $p$ for the Worker Agents to execute, and adds them to the existing Task Registry. In this section, we provide examples of some of the task text $p$ generated by the Planner Agent for each domain (molecules and peptides) during runs of PABLO. For each domain, we provide examples of 20 Planner-agent-generated tasks $p$ that led to at at least one score improvement (meaning at least one molecule generated by a Worker Agent executing the task achieved a higher score than the current best during optimization). For the molecule domain, we provide 2 examples from PABLO runs on each of the 10 GuacaMol objectives from Section 3, for a total of 20 examples.

**Molecules Example 1 - Objective: adip - Task name: CHAIN_FLEXIBILITY, Task text:**

```
Systematically modify linker chain lengths and flexibility between core
structural elements.
HINTS:
1.  Insert/remove CH_2 units in aliphatic chains.
2.  Add/remove rotatable bonds near functional groups.
3.  Test both rigid (cycloalkyl) and flexible (alkoxy) linkers.
```

**Molecules Example 2 - Objective: adip - Task name: RING_SIZE_MOD, Task text:**

```
TASK: Adjust ring sizes in high-scoring scaffolds to explore conformational
effects.
HINTS:
1.  Convert 5-membered rings to 6-membered (or vice versa).
2.  Maintain aromaticity where possible.
3.  Ensure substituents are appropriately positioned.
```

**Molecules Example 3 - Objective: fexo - Task name: BRANCHING, Task text:**

```
TASK: Increase molecular branching in hydrocarbon chains.
HINTS:
1.  Add methyl branches to aliphatic chains.
2.  Create gem-dimethyl groups.
3.  Introduce cyclopropyl rings for rigidity.
```

**Molecules Example 4 - Objective: fexo - Task name: ATOM_SWAP, Task text:**

```
TASK: Replace key carbon atoms with heteroatoms (N, O, S) in aliphatic rings
and linkers.
HINTS:
1.  Prioritize substitutions that maintain ring size but alter electronic
```

```
properties.
2.  Test bioisosteric replacements (e.g., -CH2- --> -O- in linkers).
```

**Molecules Example 5 - Objective: med1 - Task name: SPIRO_FUSE, Task text:**

```
TASK: Generate spiro-fused ring systems to explore novel conformational
constraints.
HINTS:
1.  Identify adjacent rings separated by 1-2 atoms.
2.  Merge into spiro junctions (shared single atom).
3.  Preserve peripheral substituents like isopropyl groups.
```

**Molecules Example 6 - Objective: med1 - Task name: RING_EXPANSION_V2, Task text:**

```
TASK: Expand non-aromatic rings from 5 to 6 members.
HINTS:
1.  Target rings adjacent to ketones.
2.  Use methylene insertion.
3.  Maintain bicyclic rigidity.
```

**Molecules Example 7 - Objective: med2 - Task name: QUINAZOLINONE_KETONE_SWAP_V3, Task text:**

```
TASK: Replace the quinazolinone ketone with thiazole or oxazole rings to alter
electronic distribution and hydrogen bonding.
HINTS:
1.  Maintain planarity at the core interaction site.
2.  Ensure retention of key hydrogen bond acceptors.
3.  Test both 5-membered and 6-membered heterocycle replacements.",
```

**Molecules Example 8 - Objective: med2 - Task name: RING_FUSION_ENHANCE, Task text:**

```
TASK: Generate fused polycyclic variants by merging indole with adjacent
aromatic rings through strategic bond formation.
HINTS:
1.  Create 6-5-6 tricyclic systems.
2.  Preserve indole's NH while forming new ring junctions.
3.  Explore both angular and linear fusion patterns.
```

**Molecules Example 9 - Objective: osmb - Task name: CORE_SWAP_BIOISOSTERE, Task text:**

```
TASK: Replace pyrimidine cores with bioisosteric heterocycles (e.g., triazine,
pyridone, thiazine) while preserving substituent patterns.
HINTS:
1.  Match nitrogen positioning in new cores.
2.  Maintain planar aromaticity.
3.  Evaluate both 5- and 6-membered alternative cores.
```

**Molecules Example 10 - Objective: osmb - Task name: HYDROXYL_POSITION, Task text:**

```
TASK: Systematically relocate hydroxyl groups between chain positions and ring
substituents.
```

```
HINTS:
1.  Compare terminal vs internal hydroxyl placement
2.  Test hydroxyl migration to adjacent carbons
3.  Consider diol formation in chains
```

**Molecules Example 11 - Objective: pdop - Task name: INDOLE_BRANCHING, Task text:**

```
TASK: Add alkyl or functionalized branches to indole rings in the input
molecule.
HINTS:
1.  Introduce methyl or hydroxyl groups at indole C4-C7 positions.
2.  Attach small polar groups (e.g., -CH2OH) to indole nitrogen.
3.  Preserve core indole hydrogen bonding capability.
```

**Molecules Example 12 - Objective: pdop - Task name: CHAIN_MOD, Task text:**

```
TASK: Modify alkyl chain lengths and branching in linker regions (e.g., +1/-1
CH2, add methyl branches).
HINTS:
1.  Focus on chains between amide bonds.
2.  Test both elongation and shortening.
3.  Introduce branching near aromatic systems.
```

**Molecules Example 13 - Objective: rano - Task name: FLUORINE_CHAIN_OPT, Task text:**

```
TASK: Optimize fluorinated chain geometry by adjusting double bond positions
and terminal fluorine placement.
HINTS:
1.  Shift F from terminal to penultimate position.
2.  Alternate E/Z configurations in conjugated system.
3.  Introduce cyclopropane into the chain for rigidity.
```

**Molecules Example 14 - Objective: rano - Task name: DOUBLE_BOND_MOD, Task text:**

```
TASK: Alter conjugated double bond systems.
HINTS:
1.  Shift /C=C/ positions closer to aromatic rings.
2.  Introduce second conjugated bond in chains.
3.  Test both E/Z configurations in new positions.
```

**Molecules Example 15 - Objective: siga - Task name: FLUORINE_CHAIN_RIGIDIFY_V2, Task text:**

```
TASK: Convert flexible fluorinated chains into constrained cyclopropane or
cyclobutane rings.
HINTS:
1.  Focus on chains with 3-4 carbons
2.  Preserve terminal fluorine positioning
3.  Test both geminal and vicinal fluorine patterns
```

**Molecules Example 16 - Objective: siga - Task name: FLUORINE_SYMMETRY_BREAK, Task text:**

```
TASK: Introduce asymmetry in fluorinated cyclopropane rings while preserving
total fluorine count.
HINTS:
1.  Convert symmetric CF2 groups to CF-CF2 patterns
2.  Create chiral centers through differential fluorination
3.  Test combinations of mono/di/tri-fluorinated cyclopropane rings
```

### Molecules Example 17 - Objective: valt - Task name: HYDROXYL_TUNE, Task text:

```
TASK: Modify aromatic rings by adding/removing hydroxyl groups at positions
adjacent to existing oxygen substituents.
HINTS:
1.  Prioritize positions ortho/para to ketone groups
2.  Maintain hydrogen bonding patterns
3.  Avoid over-oxidizing aliphatic regions
```

### Molecules Example 18 - Objective: valt - Task name: LACTAM_STEREOCHEMISTRY_V2, Task text:

```
TASK: Systematically invert stereochemistry at lactam alpha/beta carbons and
N-methylation sites.
HINTS:
1.  Generate all stereoisomers of top-scoring scaffolds.
2.  Test combinations of (R/S) configurations at branching points.
3.  Add/removal methyl groups at stereocenters to modulate rigidity.
```

### Molecules Example 19 - Objective: zale - Task name: SUBSTITUENT_SWAP, Task text:

```
TASK: Alter key substituents on the aromatic core while preserving overall
scaffold.
HINTS:
1.  Replace cyano groups with nitro or trifluoromethyl.
2.  Shift carboxylic acid position to meta/para locations.
3.  Introduce fluorine at unexplored positions for electronic effects.
```

### Molecules Example 20 - Objective: zale - Task name: RING_AROMATIZE, Task text:

```
TASK: Convert aliphatic rings to aromatic systems to improve rigidity and
pi-stacking potential.
HINTS:
1.  Dehydrogenate cyclohexane rings to benzene.
2.  Introduce nitrogen atoms to form pyridine-like aromatic systems.
3.  Remove saturated linkers between aromatic rings.
```

### Peptides Example 1 - Task name: LENGTH_VARIATION, Task text:

```
TASK: Optimize peptide length through truncation/extension.
HINTS:
1.  Remove 1-3 residues from N/C termini.
2.  Extend termini with K/R or hydrophobic residues.
3.  Test lengths between 15-25 amino acids.
4.  Prioritize core active region preservation.
```

**Peptides Example 2 - Task name: TERMINI_OPTIMIZE, Task text:**

```
TASK: Optimize N or C terminus modifications for improved stability and
activity.
HINTS:
1.  Add acetylation or amidation for protease resistance.
2.  Extend termini with K/R for charge enhancement.
3.  Add hydrophobic residues at termini for membrane anchoring.
4.  Test truncations to find minimal effective length.
```

**Peptides Example 3 - Task name: MOTIF_SWAPPING, Task text:**

```
TASK: Exchange motifs between high- and low-MIC peptides to identify critical
sequences.
HINTS:
1.  Replace segments in high-MIC peptides with low-MIC motifs (e.g., KLWK).
2.  Retain overall charge balance.
3.  Test shuffled motif variants.
```

**Peptides Example 4 - Task name: TRUNCATE, Task text:**

```
TASK: Identify minimal functional core sequences.
HINTS:
1.  Systematically remove 1-2 residues from N/C termini.
2.  Preserve central K/R-rich motifs and hydrophobic anchors.
3.  Test fragments for MIC retention vs.  full-length peptides.
```

**Peptides Example 5 - Task name: D_AMINO_ACID_STABILIZATION, Task text:**

```
TASK: Introduce D-amino acids to enhance proteolytic resistance without
compromising activity.
HINTS:
1.  Replace 1-3 central residues with D-amino acids.
2.  Prioritize hydrophobic residues (F/W/L) for D-substitution.
3.  Maintain N/C-terminal L-residues for binding.
4.  Test substitutions at positions 6, 9, 12.
```

**Peptides Example 6 - Task name: AROMATIC_RESIDUE_SWAP, Task text:**

```
TASK: Systematically substitute aliphatic hydrophobic residues (L/I/V) with
aromatic (F/W) in hydrophobic regions.
HINTS:
1.  Prioritize substitutions at positions 3, 7, and 11 in helical sequences.
2.  Maintain aromatic content between 15-25% of total sequence.
3.  Combine with charge redistribution for balanced amphipathicity.
```

**Peptides Example 7 - Task name: HYDROPHOBIC_FACE_BALANCE, Task text:**

```
TASK: Optimize hydrophobic face composition to prevent aggregation while
maintaining membrane affinity.
HINTS:
1.  Replace 1-2 aromatic residues with aliphatic hydrophobes (L/I/V).
```

```
2.  Create a hydrophobic face with 4-5 residues including 1-2 F/W.
3.  Ensure hydrophobic moment >0.5 for amphipathicity.",
```

**Peptides Example 8 - Task name: CHARGE_DISTRIBUTION, Task text:**

```
TASK: Optimize spatial distribution of cationic charges.
HINTS:
1.  Space K/R residues every 3-4 positions to enhance membrane interaction.
2.  Avoid charge clustering at termini; distribute along the peptide.
3.  Pair cationic residues with hydrophobic ones for synergistic effects.
4.  Test alternating charge/hydrophobic patterns.
```

**Peptides Example 9 - Task name: BETA_AMINO_ACID_INSERTION, Task text:**

```
TASK: Insert beta-amino acids to alter peptide backbone conformation and
enhance activity.
HINTS:
1.  Identify positions where beta-amino acids can improve helical structure.
2.  Insert beta-amino acids at non-core motif positions.
3.  Balance hydrophobicity and cationic charge in modified sequences.
4.  Evaluate protease resistance improvements.
```

**Peptides Example 10 - Task name: W_ENRICHMENT, Task text:**

```
TASK: Increase tryptophan (W) content in antimicrobial peptides to enhance
membrane binding.
HINTS:
1.  Substitute hydrophobic residues (L, I, V) with W in key positions.
2.  Add W at N/C termini to improve membrane interaction.
3.  Target 3-4 W residues total; avoid excessive W to prevent aggregation.
4.  Prioritize W enrichment in hydrophobic regions of amphipathic structures.
```

**Peptides Example 11 - Task name: AMPHIPATHIC_BALANCE, Task text:**

```
TASK: Improve amphipathic segregation to enhance bacterial membrane
disruption.
HINTS:
1.  Rearrange hydrophobic residues (L, I, V, F, W) to one helical face.
2.  Position cationic residues (K, R) on the opposite face.
3.  Use helical wheel projections to validate segregation.
```

**Peptides Example 12 - Task name: W_SPACING_OPTIMIZATION, Task text:**

```
TASK: Optimize spacing between tryptophan residues to enhance membrane
interaction.
HINTS:
1.  Space W residues at i, i+3, or i+4 intervals for helical alignment.
2.  Maintain 2-3 W residues in hydrophobic regions.
3.  Avoid adjacent W clusters to prevent aggregation.
```

**Peptides Example 13 - Task name: CATIONIC_REINFORCEMENT, Task text:**

```
TASK: Increase cationic charge in specific positions to enhance bacterial
membrane interaction.
HINTS:
1.  Replace neutral residues with K or R in hydrophilic regions.
2.  Add K/R at positions adjacent to hydrophobic clusters.
3.  Target a net positive charge of +4 to +6.
4.  Avoid clustering cationic residues to prevent steric hindrance.",
```

**Peptides Example 13 - Task name: INSERT_MOTIF, Task text:**

```
TASK: Insert known antimicrobial motifs into the peptide sequence.
HINTS:
1.  Try WRW, RKRK, or FLPLIG motifs.
2.  Insert at N-terminus, C-terminus, or central region.
3.  Maintain overall charge and hydrophobicity balance.
4.  Test motif orientation (forward/reverse).",
```

**Peptides Example 14 - Task name: TRUNCATE_MIDDLE, Task text:**

```
TASK: Shorten the peptide by removing residues from the middle.
HINTS:
1.  Remove 2-4 consecutive central residues.
2.  Preserve N/C-terminal motifs and charge.
3.  Verify helical structure remains intact.
4.  Test MIC after each truncation.
```

**Peptides Example 15 - Task name: AROMATIC_PATCH_ENHANCE, Task text:**

```
TASK: Increase aromatic hydrophobic residues (W/F) in clusters.
HINTS:
1.  Add W/F in positions that form aromatic patches.
2.  Maintain or enhance cationic residue distribution.
3.  Prioritize clusters in central regions of the peptide.
```

**Peptides Example 16 - Task name: WRW_MOTIF_OPTIMIZATION, Task text:**

```
TASK: Optimize WRW motif placement and surrounding residues.
HINTS:
1.  Insert WRW in hydrophobic regions.
2.  Flank with cationic residues (K/R).
3.  Maintain helical structure if applicable.
4.  Test variations in motif spacing.
```

**Peptides Example 17 - Task name: CATIONIC_CLUSTER_BOOST, Task text:**

```
TASK: Cluster cationic (K/R) and aromatic (W/F) residues to form disruptive
motifs.
HINTS:
1.  Create adjacent K/R-W/F pairs.
2.  Position clusters near hydrophobic faces.
3.  Maintain 4-6 net positive charges.
```

**Peptides Example 18 - Task name: PROLINE_INSERTION, Task text:**

```
TASK: Introduce proline residues to modulate peptide flexibility and
structure.
HINTS:
1.  Replace non-essential residues with proline at strategic positions.
2.  Insert proline at 1/3 and 2/3 sequence positions.
3.  Limit proline count to 1-2 per peptide.
4.  Test MIC impact of structural flexibility changes.
```

**Peptides Example 19 - Task name: HYDROPHOBIC_PATCH_ENHANCEMENT, Task text:**

```
TASK: Create localized hydrophobic patches while preserving cationic residues.
HINTS:
1.  Substitute neutral residues (A, S, T) with I/L/V in adjacent positions.
2.  Maintain 3-4 aromatic residues in the sequence.
3.  Avoid disrupting existing charge clusters.
```

**Peptides Example 20 - Task name: KLRWFKK_HYDROPHOBIC_OPT, Task text:**

```
TASK: Optimize hydrophobic residues within the KLRWFKK motif to enhance
membrane interaction.
HINTS:
1.  Substitute residues at positions 3, 6, and 9 with hydrophobic amino acids
(I, V, F, W).
2.  Maintain helical periodicity by preserving i, i+3, i+4 spacing.
3.  Prioritize substitutions that increase overall hydrophobic moment.
```

## L  Hand-Designed Static Worker Prompts (Ablation)

In Figure 4, we ablate the usefulness of the Planner Agent in PABLO by removing the Planner entirely and instead using a set of 10 static (unchanging) domain-specific task prompts $p_1, \ldots, p_{10}$. This ablation directly demonstrates that the dynamic modification tactics created by the Planner Agent during optimization lead to improved performance compared to just using a fixed/statics set of modification tactics throughout. In this section, we provide the exact text defining the static, domain-specific task prompts used for this ablation. The first three static task prompts, $p_1, p_2$, and $p_3$, are always exactly the three domain-specific "default" tasks used to initialize the Task Registry, which are provided in Section I.1. Below, we provide the remaining static task prompts $p_4, \ldots, p_{10}$ that we used to run this ablation on the three molecule design tasks in Figure 4:

**Ablation Static Prompt 4 ($p_4$) - Task name: ATOM_SWAP, Task text:**

```
TASK: Perform single atom swaps.
HINTS:
1.  Swap atoms in rings and linkers (e.g., C<->N, O<->S, H<->F).
2.  Keep ring count and chain length the same.
```

**Ablation Static Prompt 5 ($p_5$) - Task name: LINEARIZE, Task text:**

```
TASK: Make the molecule more flexible/linear.
HINTS:
1.  Break bulky ring systems into chains.
2.  Replace fused rings with rotatable linkers.
```

```
3.   Increase rotatable bond count.
```

**Ablation Static Prompt 6 ($p_6$) - Task name: RIGIDIFY, Task text:**

```
TASK: Make the molecule more rigid.
HINTS:
1.   Lock rotatable linkers into rings (e.g., propyl --> piperidine).
2.   Fuse aromatic rings (e.g., benzene --> naphthalene).
3.   Reduce rotatable bonds.
```

**Ablation Static Prompt 7 ($p_7$) - Task name: FUSER, Task text:**

```
TASK: Fuse adjacent rings into polycyclic cores.
HINTS:
1.   Merge rings connected by short chains (1-2 atoms).
2.   Create bicyclic or tricyclic fused systems.
3.   Preserve peripheral substituents.
```

**Ablation Static Prompt 8 ($p_8$) - Task name: REASSEMBLE, Task text:**

```
TASK: Recombine molecular fragments in new arrangements.
HINTS:
1.   Reattach functional groups at different positions.
2.   Swap positions of tail/linker/ring units.
3.   Keep approximate size, vary connectivity.
```

**Ablation Static Prompt 9 ($p_9$) - Task name: BREAK_SYMMETRY, Task text:**

```
TASK: Make the input molecule less symmetric.
HINTS:
1.   Add substituents to only one side of symmetric scaffolds.
2.   Replace one of two identical groups with something different.
3.   Vary chain lengths on symmetric branches.
```

**Ablation Static Prompt 10 ($p_{10}$) - Task name: STABILIZER, Task text:**

```
TASK: Modify the input molecule to improve general chemical stability and
reduce hazardous reactivity.
HINTS:
1.   Replace chemically unstable or hazardous motifs (e.g., peroxides,
azides/diazo, acyl halides, anhydrides, highly strained rings).
2.   Reduce overly reactive functional groups by swapping for more inert
alternatives.
3.   If already stable, propose small stabilizing variations (e.g., saturation
of labile bonds).
```

