# OpenReview forum: "Purely Agentic Black-Box Optimization for Biological Design"
_ICML.cc/2026/Conference — Submitted to ICML 2026_

### Official Review · Reviewer_a2MZ · 2026-03-12

**Soundness:** 2
**Presentation:** 3
**Significance:** 3
**Originality:** 3
**Overall Recommendation:** 4
**Confidence:** 2

**Summary:**

The paper addresses biological design as a black-box optimization problem over structured discrete spaces, such as molecules, protein and peptide sequences. To address this, the authors propose Purely Agentic Black-Box Optimization (PABLO), a hierarchical agent-based optimization framework where LLMs lead the entire search process instead of serving merely as a component within a traditional optimizer. PABLO consists of LLM-driven agents: explorer, planner and worker agents to direct exploration, strategy generation and local refinement of incumbents.
The experiments on GuacaMol standard molecule design and antimicrobial peptide optimization tasks showed that PABLO outperforms several classical and LLM-based baselines in terms of both sample efficiency and the final objective value achieved. The paper also presents ablation studies to strengthen the contributions of components of PABLO as well as optional extensions including literature tool and task awareness.

**Compliance With Llm Reviewing Policy:**

Affirmed.

**Final Justification:**

The rebuttal reinforced my prior assessment and did not provide a significant change in my evaluation, as the real-world latency and scalability concerns still remain. These should be addressed in the paper, given that it is mostly a practical method. Therefore, I will keep my score as is.

**Key Questions For Authors:**

1. How is the effectiveness of PABLO as a hierarchical multi-agent structure rather than single agent control? I mean, compared to a scientific LLM with iterative candidate refinement? Could the authors comment on this?
2. The authors emphasize token efficiency, however, how is the performance of PABLO compared to the main LLM-based baselines in terms of wall-clock runtime and approximate monetary cost per run?

**Limitations:**

yes but for the impact statement see weaknesses above.

**Strengths And Weaknesses:**

STRENGTHS

- Biological design is an important problem, and the paper addresses this challenging problem via arguing that existing structure-centric methods underuse natural-language scientific knowledge that LLMs could potentially exploit.
- The architecture of the proposed PABLO method is conceptually clear. The explorer, planner, worker decomposition is easy to understand.
- The empirical results demonstrate the strength of PABLO on various GuacaMol tasks as well as AMP optimization compared to baseline methods.
- The ablation analyses further support the high-level design choice for PABLO.


 WEAKNESSES

- The authors provide little algorithmic analysis beyond the framework description of PABLO. The contribution is clearly with an empirical system; however, it would have been more convincing if the authors provided a discussion from optimization, convergence and/or sample-complexity perspectives on replacing BO/EA-style search with an online LLM-driven strategy.
- The authors could have emphasized the practical cost and latency of PABLO more clearly in the main paper. They emphasize the token efficiency; however, Figure 2 shows that PABLO is competitive with or better than several LLM-based baselines. The token count is only a partial proxy for cost because methods differ in model sizes, inference speeds, prompt lengths, and sequential interaction structure. There is discussion provided in Appendix C; however, since these limitations are real because agentic systems are mostly bottlenecked by long sequential LLM interactions, it could have been more visible in the main text.
- I believe the purely agentic is overly framed in the paper. Indeed, the search is driven by LLM agents; however, there are still some decisions that require manual adjustments, such as diversity thresholds and seed selection, which makes PABLO more like a carefully engineered LLM-agent-based system rather than a pure unconstrained optimizer.
- The impact statement is poorly written for an ICML-level quality paper. Saying that there are many possible societal consequences but none need to be highlighted, is not adequate for a paper on biological design and antimicrobial discovery.

MINOR

- Spacing after Figures and Tables should be according to the ICML template style. (spacing after Table 1 and Figure 2 caption).

---

> ### Author Rebuttal · Authors · 2026-03-30
>
> Thank you for the thoughtful review!
> ### Q1
> The **"PABLO w/o Both (3 Static)"** variant in Fig. 4 provides a direct comparison to a single-agent setup: it removes both the Explorer and Planner, leaving only a single Worker agent performing iterative candidate refinement. This variant consistently underperforms PABLO.
>
> We note that the Worker agent in this ablation uses gpt-oss-120b, a faster, less expensive model suited to the high-volume local refinement step. Running a single-agent baseline with Intern-S1 (the larger, more capable model used for Explorer/Planner) would be computationally infeasible — a single Intern-S1-only run to 20K evaluations would take over **2 weeks** to complete. Part of why the multi-agent hierarchy works well in practice is precisely this division of labor: the more expensive model (Intern-S1) handles the **less frequent** but more sophisticated tasks (global exploration, strategy synthesis), while the faster model handles the **high-volume** local refinement. This design is itself a benefit of the hierarchical decomposition.
> ### Q2/W2
> We agree that token count alone may not tell the complete story for the practical cost. We plan to add the following table to the main text:
> | | $/run | Hours/run |
> |--------|-----------|-----------|
> | PABLO | $8 | 100 |
> | LLAMBO | $157 | 100 |
> | AlphaEvolve | $0.19 | 1 |
> | BOPRO-L | $71 | 200 |
> | BOPRO-S | $2.36 | 50 |
> | MOLLEO | $20 | 50 |
> Note that cost estimates above are approximations using token counts from Figure 2, each baseline's LLM, and public API pricing. Total costs of most methods (with the exception of BOPRO-L and LLAMBO) are generally comparable, with the notable exception of AlphaEvolve performing extremely well on token efficiency by modifying a program rather than having agents do decision making. We will highlight this strength of AlphaEvolve specifically.
> ### W1
> We absolutely agree with you in principle, but a believable analysis is just very hard for these kinds of structured optimization problems regardless of whether you are using agents, BO, or any other framework. BO convergence results for example typically (1) assume the objective is a sample path from your chosen prior (or roughly equivalent frequentist assumptions about bounded RKHS norm), and (2) result in extremely pessimistic regret bounds in the dimensionality of the objective function. (1) is a highly suspicious assumption in the structured domain in particular, because it would require believing e.g. that the antimicrobial activity of a peptide is a draw from a Gaussian process whose prior covariance function is an RBF kernel on top of a massive normalizing flow encoder. (2) has certainly become more interesting in recent years as e.g. local and smoothness-assuming approaches to BO achieve sharply more optimistic results than the bounds imply.
>
> For studying LLMs formally as optimizers specifically, there are serious roadblocks. For example, it is known that simply swapping the order of instructions to the LLM changes the output token probabilities, even if the text is identical. This non-exchangeability of conditioning on information makes LLMs fundamentally incompatible with traditional analyses of posterior inference one might like to do, even before coming to the problem that conditioning the LLM densely on the input space – a fundamental requirement for eventual convergence in other optimization schemes is to eventually collapse uncertainty everywhere – seems challenging with in context learning alone due to context sizes.
> ### W3
> We agree with this nuance. By "purely agentic," we mean that **the search policy itself is driven by LLM agents** rather than by BO acquisition functions, surrogate fitting, or EA operators. We do not mean the system has zero implementation choices. Diversity thresholds, validity filters, etc. are **auxiliary control mechanisms** around an agent-driven search core, analogous to how standard BO algorithms still require engineering choices around surrogate model, acquisition function, etc. We will update the text to clarify this distinction. Maybe “purely agent-driven” would be a more reasonable title to emphasize that the optimization process is driven entirely by agents but not that all decisions are made by LLMs?
> ### W4
> We will use the following impact statement:
> *"This work presents methods for computationally guided biological design, with a specific application to antimicrobial peptide discovery. The beneficial potential is significant: antimicrobial resistance is a growing global health crisis, and accelerating the discovery of novel antimicrobial agents could save lives. However, generative biological design tools also carry dual-use risks: the same methods that can design beneficial therapeutics could in principle be misused to engineer harmful biological agents. We believe responsible deployment of such systems requires institutional oversight and strict adherence to biosafety protocols."*
> ### Minor
> We will fix spacing.

---

> > ### Author Rebuttal · Reviewer_a2MZ · 2026-04-01
> >
> > The rebuttal clears up several presentation points and improves the overall clarity of the paper, including the motivation behind the hierarchical design, the wording around “purely agentic,” and the impact statement, which are all better now. The cost table is also a helpful step toward making things more transparent.
> >
> > That said, my main concern about the lack of algorithmic or optimization-level insight is still largely unresolved. The response mostly explains why such analysis is difficult, rather than offering any concrete intuition or perspective. Also, while the cost estimates are useful, the discussion of real-world latency and scalability still feels incomplete.
> >
> > Overall, I would consider my concerns partially addressed and keep my score.

---

> > > ### Author Response · Authors · 2026-04-02
> > >
> > > Thank you for the thoughtful follow-up and for acknowledging the improvements to the hierarchical design motivation, "purely agentic" framing, impact statement, and cost table. We are glad these clarifications were helpful.
> > >
> > > We address your two remaining concerns below.
> > >
> > > ## Algorithmic / Optimization-Level Insight (W1)
> > >
> > > We appreciate this concern and want to offer some additional concrete perspective beyond "it's difficult." At a high level, PABLO can be understood as an **online policy over a combinatorial space of natural-language search strategies**, where the black-box objective provides a reward signal that drives strategy selection. The Planner maintains a task registry with empirical success rates, effectively implementing a **multi-armed bandit over local search operators** — strategies that consistently improve candidates are selected more often, while unproductive ones are pruned. The Explorer provides a complementary **global diversification** mechanism, analogous to the restart/exploration component in algorithms like random-restart hill climbing or CMA-ES population resampling. The key algorithmic insight is that expressing both local strategies and global hypotheses in natural language — rather than as fixed mutation operators or acquisition functions — allows the system to **adapt its search policy online** based on observed structure-activity patterns, which fixed-strategy optimizers cannot do.
> > >
> > > That said, we fully agree that formalizing this intuition into convergence or sample-complexity guarantees remains an open problem. As we discussed in the rebuttal, this is non-trivial for *any* optimizer over structured discrete spaces. We further agree with a perhaps implicit point here that methods like BO have storied histories of intuition about when and how they work that agents lack. We just feel that, in many cases the ability of an agent to (1) condition on prior knowledge baked into the model parameters or a searchable database, and (2) condition on feedback written in natural language–as evidenced by the relatively consistent lift obtained by the use of a literature search tool–is a fairly significant advantage even if there is a lot left to figure out.
> > >
> > > ## Real-World Latency and Scalability (Q2/W2)
> > >
> > > We agree that the cost comparison table provided in our rebuttal should be accompanied by significant discussion in the main text of real-world latency, scalability, and practical trade-offs. We will include this in the camera-ready version. Most obviously, as we discuss already briefly in the paper but will expand, AlphaEvolve is quite token efficient even if the final optimization performance is often somewhat worse, which is a key trade-off in certain situations. We generally think that the total price differences (e.g., \\$8 vs \\$0.19) are not *that* large in situations where you are running a reasonable amount of optimization or where the optimization objective itself incurs costs. For example, even when using in silico oracles, the compute time cost of running the objective function can dwarf either of these in reserved GPU time compared to using tokens as a service. With that said, in situations where many or repeated optimization runs are called for, the cost savings could be worth it and users should do a trade-off calculation for themselves.

---

### Official Review · Reviewer_uZPY · 2026-03-13

**Soundness:** 3
**Presentation:** 3
**Significance:** 3
**Originality:** 3
**Overall Recommendation:** 4
**Confidence:** 2

**Summary:**

The paper studies biological black-box optimization over structured design spaces and frames it as a fully agentic, language-based reasoning problem rather than using an LLM as a narrow component within a conventional optimizer. Specifically, the authors propose PABLO, a hierarchical agentic framework that compresses the optimization history into a global context and decomposes search into an explorer for global candidate generation, a planner for generating and selecting natural-language local search tasks via a task registry, and worker agents for iterative local refinement.

**Compliance With Llm Reviewing Policy:**

Affirmed.

**Final Justification:**

The rebuttal addressed my concern. After reading other reviewers' comments, I agree that the real-world latency and scalability concerns still exist.

**Key Questions For Authors:**

1. Could you further disentangle which parts of the gain come from the hierarchical agentic decomposition itself, versus the choice of a strong science-pretrained LLM and the downstream validity/deduplication/constraint-handling pipeline? Additional ablations that progressively simplify the explorer/planner/worker stack would make it easier to understand where the main improvements are coming from.

2. Since the paper positions PABLO as a domain-general framework for biological black-box optimization, can the authors clarify which components would need to be redesigned when moving beyond molecules and AMPs (e.g., to protein design)? It would be helpful to know how much domain-specific effort is required to define the task registry, default hints, validity filters, and local search actions in a new domain.

**Limitations:**

yes

**Strengths And Weaknesses:**

Strength:

1. The paper presents a clear and timely reframing of biological black-box optimization as a fully agentic, language-based reasoning problem. The explorer/planner/worker decomposition is clean and different from prior pipelines that only use an LLM as a subroutine.


2. The method is designed for realistic design settings rather than only benchmark optimization. In particular, the task registry, validity/novelty filtering, hard-constraint handling, and optional literature-retrieval extension make the framework appealing for practical biological design workflows.


Weakness

1. The empirical scope does not fully match the paper’s broad domain-general claims. While the introduction motivates applications including protein engineering, the experiments are concentrated on molecule optimization and antimicrobial peptides, so generalization to broader biological modalities remains somewhat speculative.


2. The “purely agentic” framing relies on substantial hand-crafted, domain-specific scaffolding. The method initializes domain-specific default tasks with manually written hints and uses a bounded task registry, which makes the system less automatic than the headline may initially suggest and raises questions about portability to new domains.


3. It is unclear which part of the performance gain comes from the agentic decomposition itself versus prompt engineering, strong scientific LLM priors, and deterministic post-processing. The ablations on planner/explorer are useful, but they do not fully disentangle these factors.

4. The empirical positioning against recent LLM-based molecular optimizers is still somewhat incomplete. While the paper compares against several strong baselines, including LLAMBO, AlphaEvolve, BOPRO, and MOLLEO, it does not discuss or compare to LICO [1], which is a closely related LLM-based black-box molecular optimization method evaluated on PMO. Also, SAGA [2] seems more relevant as conceptually related agentic work, which studies goal/objective evolution rather than fixed-objective optimization.


[1] Nguyen, Tung, and Aditya Grover. "Lico: Large language models for in-context molecular optimization." ICLR 2025.

[2] Du, Yuanqi, et al. "Accelerating Scientific Discovery with Autonomous Goal-evolving Agents." arXiv preprint arXiv:2512.21782 (2025).

---

> ### Author Rebuttal · Authors · 2026-03-31
>
> Thank you for your feedback! We address each concern below.
> ### Q1/W3
> - **Choice of LLM:** Re-running full optimization sweeps with each possible LLM variant is extremely compute-intensive and outside the scope of this work. We view systematic study of how LLM choice affects agentic optimization as an important direction for future work. Please see our response to **R4 Q1**.
> - **Validity/constraint-handling:** These are lightweight, standard-practice checks — e.g., verifying that a proposed peptide contains only valid amino acid characters and skipping candidates already evaluated. These checks are inexpensive safeguards that any practitioner would apply before querying a potentially costly black-box oracle (e.g., wet-lab synthesis). Removing them would waste evaluation budget on invalid or redundant candidates, reducing sample efficiency without providing meaningful signal about the agentic decomposition.
> - **Progressively simplifying agent stack:** We believe Fig. 4 already provides this ablation. It progressively removes components while holding all other factors constant (same LLM, same post-processing pipeline, etc.): "PABLO w/o Explorer" removes global search; "PABLO w/o Planner (3 Static)" and "(10 Static)" replace dynamic planning with fixed prompts; "PABLO w/o Both (3 Static)" removes both Explorer and Planner, reducing the system to a single Worker agent iterating with fixed prompts — essentially the single-agent iterative refinement baseline. Performance degrades consistently at each simplification step, directly attributing gains to the hierarchical decomposition. We suspect there may have been some confusion here and will revise the text to improve clarity. If the reviewer has specific additional simplifications in mind beyond those in Fig. 4, we would welcome the suggestion and are happy to discuss further!
> ### Q2/W1
> We agree the framing can be sharpened. Our experiments cover **two distinct biological modalities** — small molecules and peptides — which differ substantially in representation, search space, and objective structure. Transitioning between molecules and peptides required changing only the validity filter and 3 default task hints; the core optimization loop was unchanged. Please see our response to **R1 Q3** for more on applying PABLO to new domains.
> ### W2
> By "purely agentic," we mean that **the search policy itself is driven by LLM agents** rather than by BO acquisition functions, surrogate fitting, EA operators or RL. We do not mean the system has zero implementation choices. Diversity thresholds, validity filters, etc. are **auxiliary control mechanisms** around an agent-driven search core, analogous to how standard BO algorithms still require engineering choices around the surrogate, acquisition function, etc. Arguably BO–at least when using a latent space like NF-BO and ApexGO do to accomplish structured optimization–requires domain-specific hand tailoring because you must collect valid examples and train a bespoke generative model in that domain that exposes a latent space you can explore; you would still want to filter the outputs of the BO generative model down to valid outputs. Regardless: maybe “purely agent-driven” would be a more reasonable title to emphasize that the optimization process is driven entirely by agents but not that all decisions are made by LLMs? Based on your feedback and another reviewer with the same question, we would be happy to make this change. Also, please see **R1 Q3** for more on domain-specific components.
> ### W4
> We agree that both LICO and SAGA are relevant and we plan to add discussion of each. LICO reports **AUC of top-10 average** (PMO metric). To enable direct comparison, we computed PABLO's AUC top-10 from our raw optimization trajectories. Below we compare against LICO's Table 1 (1K budget) and Table 2 (10K budget) on all 10 GuacaMol tasks:
>
> **AUC Top-10 @ 1K**:
> |Task|PABLO|LICO|Diff|
> |------|-------|------|------|
> |adip|0.578 ± 0.014|0.541 ± 0.026|+0.037|
> |fexo|0.764 ± 0.022|0.700 ± 0.023|+0.064|
> |med1| 0.266 ± 0.012|0.217 ± 0.019|+0.049|
> |med2| 0.259 ± 0.014|0.193 ± 0.009|+0.066|
> |osmb| 0.845 ± 0.015|0.759 ± 0.008|+0.086|
> |pdop| 0.598 ± 0.014|0.473 ± 0.009|+0.125|
> |rano| 0.837 ± 0.018|0.687 ± 0.029|+0.150|
> |siga| 0.457 ± 0.048|0.315 ± 0.097|+0.142|
> |valt| 0.114 ± 0.161|0.000 ± 0.000|+0.114|
> |zale| 0.476 ± 0.011|0.404 ± 0.022|+0.072|
> | **Sum**|**5.194**|**4.289**|**+0.905**|
> **AUC Top-10 @ 10K**:
> |adip|0.686 ± 0.020|0.679 ± 0.027|+0.007|
> |fexo|0.863 ± 0.033|0.772 ± 0.023|+0.091|
> |med1|0.370 ± 0.011|0.291 ± 0.016|+0.079|
> |med2|0.320 ± 0.015|0.280 ± 0.019|+0.040|
> |osmb|0.896 ± 0.008|0.820 ± 0.012|+0.076|
> |pdop|0.724 ± 0.028|0.557 ± 0.028|+0.167|
> |rano|0.886 ± 0.014|0.774 ± 0.008|+0.112|
> |siga|0.718 ± 0.041|0.567 ± 0.034|+0.151|
> |valt|0.709 ± 0.217|0.000 ± 0.000|+0.709|
> |zale|0.567 ± 0.026|0.515 ± 0.017|+0.052|
> |**Sum**|**6.739**|**5.255**|**+1.484**|
> PABLO outperforms LICO on **10/10 tasks at both budgets**. We will add these results.

---

> > ### Author Rebuttal · Reviewer_uZPY · 2026-04-04
> >
> > Thanks for your response. My concerns are addressed. I will adjust my score accordingly.

---

> > > ### Author Response · Authors · 2026-04-06
> > >
> > > Great to hear, thanks again for your feedback!

---

### Official Review · Reviewer_Ai7R · 2026-03-13

**Soundness:** 3
**Presentation:** 3
**Significance:** 3
**Originality:** 3
**Overall Recommendation:** 5
**Confidence:** 3

**Summary:**

This paper introduces PABLO (Purely Agentic Black-box Optimization), a framework driven purely by large language model (LLM) agents for tackling black-box optimization problems in biological design. Unlike existing approaches that embed LLMs as sub-modules within Bayesian optimization or genetic algorithms, PABLO orchestrates multiple LLM agents—Planner, Worker, and Explorer—to collaboratively drive the entire optimization loop using natural language. This includes generating new hypotheses, planning search strategies, and performing local refinements. The method is evaluated on tasks involving antimicrobial peptide design and GuacaMol molecular optimization, where it consistently outperforms both traditional methods and current LLM-augmented baselines in terms of sample efficiency and final solution quality. Additionally, the antimicrobial activity of peptides designed by PABLO is validated through in vitro experiments.

**Compliance With Llm Reviewing Policy:**

Affirmed.

**Key Questions For Authors:**

See Weaknesses

**Limitations:**

yes

**Strengths And Weaknesses:**

* Strengths

(1) The paper is well-designed and presents thorough experimentation. It includes comprehensive comparisons across multiple benchmarks and validates the effectiveness of its key components through ablation studies.

(2) The paper is clearly structured, with a logical and coherent narrative that flows seamlessly from motivation to methodology and experimental results.

(3) This work explores a cutting-edge application of LLMs in scientific discovery, demonstrating strong potential for practical impact.

* Weaknesses

(1) The proposed method heavily relies on the reasoning capabilities of the underlying LLM, yet this dependency appears to be insufficiently explored or discussed.

(2) The paper does not appear to include an analysis of the reasoning reliability of the LLM.

---

> ### Author Rebuttal · Authors · 2026-03-30
>
> We thank the reviewer for the positive assessment and for recognizing the thorough experimentation, clear presentation, and practical impact of our work. We address both of your questions below.
>
>
> ### LLM Reasoning Dependency (Q1)
>
> This dependency is indeed central to PABLO, and we agree it deserves more explicit discussion. Importantly, PABLO does not rely on unconstrained free-form reasoning. The planner/explorer/worker decomposition is specifically designed to structure and constrain reasoning into an optimization loop: the Explorer proposes diverse global candidates, the Planner converts history into explicit task-level strategies, and the Worker performs bounded, score-driven local refinement. The feedback from the black-box objective function selects for useful/good reasoning over time, regardless of whether any individual reasoning step is "correct". We are happy to discuss this with you further.
>
> ### Reasoning Reliability (Q2)
>
> Our current evidence for reasoning reliability is indirect but substantive: the ablations (Figure 4) show that removing the Planner or Explorer consistently degrades performance, indicating that the reasoning these agents contribute is functionally useful rather than decorative. At the same time, not every proposed task or refinement step is productive — this is precisely why PABLO uses iterative score-based selection and the MAX_FAILS mechanism to abandon unproductive strategies. PABLO's architecture is robust to imperfect reasoning because the objective function scores provide a grounding signal that selects for productive reasoning over time. Purely for the goal of blackbox optimization, the objective function serving as a verifiable reward signal at the macro level renders incorrect reasoning at the micro (e.g. per step) level irrelevant, because incorrect reasoning translates to slower optimization progress. With that said, if we want to make claims about LLMs improving the interpretability of the optimization process and decision making (which we do not in this paper and will clarify) beyond what is possible with traditional methods, then we absolutely agree with your point here: reliable reasoning at the per-step level would become mandatory. It would be extremely interesting in future work to, for example, correlate a human judge’s opinion of the planner’s reasoning with the success of each step.
>
> Thank you again for the encouraging assessment!

---

> > ### Author Rebuttal · Reviewer_Ai7R · 2026-04-07
> >
> > Overall, I would consider my concerns addressed and keep my score.

---

### Official Review · Reviewer_eZtF · 2026-03-13

**Soundness:** 3
**Presentation:** 3
**Significance:** 3
**Originality:** 3
**Overall Recommendation:** 4
**Confidence:** 2

**Summary:**

This paper replaces traditional structure-centric optimization in molecular and biological sequence design with a language-driven, multi-agent reasoning framework. The proposed system, PABLO, utilizes a hierarchy of Explorer, Planner, and Worker agents to handle global search, localized tactics, and candidate refinement. Its main contribution lies in demonstrating that a purely agentic approach—operating without standard mathematical acquisition functions or evolutionary scaffolds—can reach state-of-the-art performance on GuacaMol and antimicrobial peptide (AMP) tasks.

PABLO’s architecture accommodates semantic constraints and literature retrieval tools with ease. The methodology’s flexibility is a significant strength, and the authors provide evidence of its utility through in vitro validation. These experiments confirm that the diverse peptide portfolios designed by PABLO exhibit strong inhibitory activity against clinical bacterial pathogens. Furthermore, the system maintains high sample efficiency and competitive token consumption throughout the design process.

**Compliance With Llm Reviewing Policy:**

Affirmed.

**Key Questions For Authors:**

1.The foundation models used, such as Intern-S1, are pretrained on vast chemical and biological datasets. This raises a critical question: are the strong GuacaMol results a product of the agentic optimization process or simply the retrieval of memorized high-scoring molecules? Clarifying this distinction is necessary. Providing evidence or a stronger argument against data contamination would significantly increase my confidence in the benchmark comparisons.

2. As noted in Appendix D.2, PABLO resorts to random sampling to find a non-zero score when initialized in a zero-signal regime. This approach seems fragile. If random sampling fails to find a non-zero score in a highly sparse reward landscape, how would the framework adapt or explore without an initial gradient or reasoning signal? I am interested in how the system handles cases where the "cold start" problem persists beyond the initial samples.

3. While the Planner generates tasks dynamically, the system remains anchored by hand-crafted system prompts, including specific prefixes, suffixes, and literature-based templates. Adapting PABLO to a completely different biological modality, such as RNA, DNA, or larger proteins, appears to require non-trivial manual effort. The authors should clarify the amount of domain-specific prompt engineering required for such transitions.

4. The 100-hour wall-clock time and high token usage are significant practical bottlenecks. Have the authors experimented with using smaller, more efficient open-weight models for the Explorer and Planner roles to reduce these overheads? If such experiments were conducted, it is important to know the extent of the performance degradation compared to the current configuration.

**Limitations:**

yes

**Strengths And Weaknesses:**

### Strengths

This work replaces standard mathematical scaffolds, such as BO or EAs, with a hierarchical LLM reasoning framework. The PABLO architecture—comprising Explorer, Planner, and Worker agents—integrates natural language constraints and RAG directly into the optimization loop. Performance is strong; the method reaches state-of-the-art levels on the GuacaMol benchmark and outperforms baselines in antimicrobial peptide (AMP) design.

A standout feature is the *in vitro* validation. By synthesizing and testing 20 optimized peptides against bacterial pathogens, the authors provide evidence of real-world therapeutic utility that purely *in silico* studies often lack. The ablation studies are also thorough, specifically regarding the necessity of each agent and the impact of dynamic prompting. The authors are transparent about token usage and the computational costs associated with their approach.

### Weaknesses

Efficiency is a significant hurdle. Wall-clock times reach approximately 100 hours for 20K evaluations, making the method far slower and more expensive than traditional, lightweight structure-centric algorithms. There is also the unresolved risk of data contamination. Because models like Intern-S1 are trained on extensive scientific literature, it is difficult to isolate the framework's reasoning capabilities from its latent knowledge of the GuacaMol benchmark.

The system also falters during initialization in flat landscapes. As noted in Appendix D.2, PABLO requires random sampling to escape zero-signal regions where the LLM has no feedback to act upon. This suggests the reasoning process is not yet a complete replacement for stochastic search in the early stages of optimization. Finally, the framework relies on specialized domain-specific meta-prompts and heuristic hints. This dependence indicates that applying PABLO to new modalities will likely require substantial manual prompt tuning rather than being an automated, out-of-the-box solution.

---

> ### Author Rebuttal · Authors · 2026-03-30
>
> Thank you for the thoughtful review!
> ### Q1
> We want to emphasize several points that argue against simple memorization as the explanation for PABLO's strong empirical performance:
> 1. **PABLO operates as a black-box optimizer** that receives only SMILES strings and numeric scores. It is not given target molecules, oracle definitions, or task names. In our primary GuacaMol results (Figure 3, Table 1), we do **not** use task descriptions or task awareness — the agents see only structure-score pairs.
> 2. **Optimization trajectories show gradual improvement** over thousands of evaluations rather than immediate recovery of near-optimal molecules. This is inconsistent with simple retrieval of memorized optima.
>
> We will strengthen this discussion and acknowledge data contamination as an inherent caveat for any LLM-based optimization approach.
> ### Q2
> The random sampling fallback is needed only for the rare edge-case when the initialization dataset contains no score variation (*truly zero-variance** initialization). Importantly, when there is even minimal score differentiation (e.g., scores of 0.0 vs. 1e-100), PABLO can leverage this signal effectively. In our experiments, this edge-case arose for only one objective (valsartan SMARTS). In such settings, no optimizer has useful information to exploit. Random initialization is in fact standard practice across the black-box optimization and BO literature. For example, most standard BO methods begin with a random or space-filling initialization phase (e.g., LHS, Sobol) before switching to acquisition-guided search (e.g., [1,2]). PABLO's fallback to random sampling in this regime is consistent with this standard practice.
> We agree with the reviewer that PABLO  — like other black-box optimization methods  — is NOT a “complete replacement for stochastic search in early stages of optimization”. We do not claim that PABLO eliminates all stochastic search, but rather, that it replaces the **main optimization policy** with language-based reasoning. We will clarify this distinction.
> [1] Garnett, R. Bayesian Optimization. Cambridge University Press, 2023.
> [2] Balandat et al. BoTorch: Bayesian Optimization in PyTorch, software library, 2020.
> ### Q3
> We agree PABLO is not zero-effort to apply to new domains, and we will clarify this in the camera-ready version.
> The core optimization loop — planner/explorer/worker decomposition, task registry dynamics, history compression, score-driven adaptation, etc. — remains unchanged across domains. Furthermore, we made significant effort to define core PABLO agents and prompts in a domain-agnostic way so that adding new domains is as easy as possible.
> Applying PABLO to a new domain (e.g., DNA, RNA, larger proteins) requires updating **two domain-specific components**:
> 1. **Validity filter:** An inexpensive function checking that a proposed candidate is a valid object in the relevant search space. For molecules, this is a check that a SMILES string represents a valid structure (via RDKit). For peptides, it is a check that the sequence contains only allowed amino acid characters. Applying this to DNA is as simple as checking that the sequence contains only A, C, G, and T.
> 2. **Default task hints:** PABLO bootstraps the task registry with 3 default tasks that serve as examples to ground the Planner in what a reasonable "modification" looks like. Importantly, these were **themselves generated by prompting an LLM** (e.g., "suggest 3 simple ways to modify a molecule") — no domain expertise or manual prompt engineering was required. Adapting to a new domain is as simple as prompting an LLM with the new domain name (e.g., "suggest 3 simple ways to modify a DNA sequence"). We will add further discussion of this.
>
> Outside of these, all other prompts stay the same except for literal domain names (e.g., find-and-replace "SMILES string" with "DNA sequence"). We also note that our provided GitHub codebase includes a **step-by-step guide in the README** for running PABLO on any new domain.
> ### Q4
> We agree that runtime deserves more prominence. The ~100-hour wall-clock time is a real practical limitation. Systematically ablating LLM choice is an important direction but outside scope due to the compute cost of full optimization sweeps per model variant. We will foreground this tradeoff in the main text and clearly separate **sample efficiency** (where PABLO excels) from **wall-clock efficiency** (where some classical methods may remain faster). In realistic biological design, oracle cost (e.g., wet-lab experiments) typically dominates compute cost, making sample efficiency the more relevant metric.
> Furthermore, classical (non-LLM) ML methods for biological design often rely on computationally expensive deep generative models to generate candidate molecules or proteins. As a result, these approaches often incur substantial wall-clock runtimes comparable to LLM-based methods. For example, NF-BO requires approximately 50 hours on the same GuacaMol tasks.

---

### Decision · Program_Chairs · 2026-04-30

**Decision:**

Reject

**Comment:**

This paper introduces PABLO, a purely agentic black-box optimization framework for biological design by replacing traditional structure-centric optimizers with a hierarchy of LLM agents system. This system uses pretrained LLMs to drive global search, strategy selection, and local refinement. This method achieves state-of-the-art performance on GuacaMol molecular optimization and antimicrobial peptide (AMP) design tasks, supported by in vitro validation demonstrating real-world inhibitory activity.

**Strengths:**

- Clean hierarchical agentic decomposition distinct from prior LLM-as-subroutine approaches. (eZtF, uZPY, a2MZ).
- Strong empirical results on GuacaMol and AMP tasks, with thorough ablation studies validating each component. (eZtF, Ai7R, uZPY, a2MZ)


**Weaknesses:**
- Limited algorithmic or optimization-theoretic insight.
- High wall-clock time (~100 hours) limits practical scalability. (eZtF, a2MZ)
- Data contamination risk from LLM pretraining on scientific literature. (eZtF)
- "Purely agentic" framing overstated; system requires manual prompts, thresholds, and filters. (eZtF, uZPY, a2MZ)
- Lack of algorithmic convergence or sample-complexity analysis. (a2MZ)
- Missing comparisons to LICO and SAGA (added in rebuttal). (uZPY)

**Additional Comments On Reviewer Discussion:**

Reviewers' concerns (Ai7R,uZPY) are mainly addressed in the rebuttal. a2MZ noted algorithmic insight and real-world scalability issue remain partially unresolved.

Overall, the rebuttal concedes that formal analysis is intractable. In the context of ICML's high standards, this places the paper in a weakly-grounded category without sufficient theoretical guidance. I suggest reject of the paper.